# Cell cycle length governs heterochromatin reprogramming during early development in non-mammalian vertebrates

Hiroto S Fukushima [ID][1,2 ✉], Takafumi Ikeda[1,3,4], Shinra Ikeda[1] & Hiroyuki Takeda [ID][1,4 ✉]

## Abstract

Heterochromatin marks such as H3K9me3 undergo global erasure and re-establishment after fertilization, and the proper reprogramming of H3K9me3 is essential for early development. Despite the widely conserved dynamics of heterochromatin reprogramming in invertebrates and non-mammalian vertebrates, previous studies have shown that the underlying mechanisms may differ between species. Here, we investigate the molecular mechanism of H3K9me3 dynamics in medaka (Japanese killifish, *Oryzias latipes*) as a non-mammalian vertebrate model, and show that rapid cell cycle during cleavage stages causes DNA replication-dependent passive erasure of H3K9me3. We also find that cell cycle slowing, toward the mid-blastula transition, permits increasing nuclear accumulation of H3K9me3 histone methyltransferase Setdb1, leading to the onset of H3K9me3 re-accumulation. We further demonstrate that cell cycle length in early development also governs H3K9me3 reprogramming in zebrafish and *Xenopus laevis*. Together with the previous studies in invertebrates, we propose that a cell cycle length-dependent mechanism for both global erasure and re-accumulation of H3K9me3 is conserved among rapid-cleavage species of non-mammalian vertebrates and invertebrates such as *Drosophila, C. elegans, Xenopus* and teleost fish.

Keywords Development; Epigenome; Heterochromatin; Mid-blastula Transition; Reprogramming
Subject Categories Chromatin, Transcription & Genomics; Development

## Introduction

Heterochromatin represses gene expression by physical folding and serves as epigenetic memory in cells, and the repressive histone modification, H3K9me3, plays a central role in the establishment and maintenance of heterochromatin (Allshire and Madhani, 2018; Padeken et al, 2022). After fertilization, cellular memories installed during gametogenesis undergo reprogramming, which allows embryonic cells to acquire pluripotency (Xia and Xie, 2020; Xu and Xie, 2018). Indeed, during this process, heterochromatin marks such as H3K9me3 are erased and re-installed (Mutlu et al, 2018, 2019; Seller et al, 2019; Laue et al, 2019; Fukushima et al, 2023; Bessler et al, 2010; Yuan and O'Farrell, 2016; Hontelez et al, 2015; Wang et al, 2018; Yu et al, 2022; Zhou et al, 2023). The proper reprogramming of H3K9me3 is essential for normal development. For example, previous studies of somatic cell nuclear transfer suggest that heterochromatin marks, if remain during early development, act as an epigenetic barrier to entry into normal development (Matoba et al, 2014; Chung et al, 2015; Jullien et al, 2017; Xu et al, 2023), and experimental removal of H3K9me3 leads to abnormal development (Fukushima et al, 2023). However, in spite of well-recognized dynamics and importance of heterochromatin reprogramming in early development, how this process is regulated remain elusive, in particular, in vertebrates.

In differentiated cells, H3K9me3 is deposited by histone methyltransferases Suv39h1/2 and Setdb1, and erased actively by enzymes (e.g., demethylase) or passively by DNA-replication-coupled dilution of histone modifications (Allshire and Madhani, 2018; Padeken et al, 2022). Although the timing and extent of heterochromatin reprogramming vary between species, they should be tightly regulated during early development because of their profound impacts on transcription and genome structure (Allshire and Madhani, 2018; Padeken et al, 2022). The vast majority of animals begin development with rapid cell divisions and exhibit the conserved dynamics of transcription and epigenetic reprogramming. Mammalian embryos, however, are characterized by a slow cell cycle, an early onset of transcription, and unique epigenetic reprogramming (Tadros and Lipshitz, 2009; Vastenhouw et al, 2019; Xia and Xie, 2020). In rapid-cleavage species such as *Drosophila, C. elegans*, and non-mammalian model vertebrates (Tadros and Lipshitz, 2009; Vastenhouw et al, 2019), H3K9 methylations are detected in germ cells, almost undetectable after fertilization, and detected again after several rounds of cleavage, indicating that H3K9me3 is subject to global erasure and re-accumulation (Mutlu et al, 2018, 2019; Seller et al, 2019; Laue et al, 2019; Fukushima et al, 2023; Bessler et al, 2010; Yuan and O'Farrell, 2016; Hontelez et al, 2015). In those species, the slowing of the cell cycle and zygotic genome activation (ZGA) coincide with the re-

[1]Department of Biological Sciences, Graduate School of Science, The University of Tokyo, Tokyo 113-0033, Japan. [2]Present address: Center for Integrative Medical Sciences, RIKEN, Yokohama 230-0045, Japan. [3]Present address: Institute for Protein Dynamics, Kyoto Sangyo University, Kyoto 603-8555, Japan. [4]Present address: Faculty of Life Sciences, Kyoto Sangyo University, Kyoto 603-8555, Japan. ✉E-mail: hiroto.fukushima@riken.jp; takeda_h@cc.kyoto-su.ac.jp

establishment of heterochromatin (Tadros and Lipshitz, 2009; Vastenhouw et al, 2019), raising the possibility that the two events are involved in the onset of H3K9me3 deposition in early development. However, despite the highly conserved dynamics of heterochromatin reprogramming among rapid-cleavage species, species-specific mechanisms have been proposed. In *Drosophila* and *C. elegans*, cell cycle slowing is a major mechanism that times the onset of H3K9me3 deposition by allowing nuclear accumulation of histone methyltransferase (Mutlu et al, 2018, 2019; Seller et al, 2019). On the other hand, Laue et al, demonstrated in zebrafish that ZGA-dependent clearance of the maternally provided chromatin remodeler, Smarca2, is a prerequisite for re-accumulation of H3K9me3 (Laue et al, 2019). Therefore, the underlying mechanism varies even between these rapid cleavage species. Furthermore, the mechanism that drives the erasure process is even less understood.

In this study, we address what drives the erasure of heterochromatin during cleavage and what determines the timing of heterochromatin re-establishment, mainly in medaka, Japanese killifish (*Oryzias latipes*). The medaka is another good non-mammalian model organism with a large evolutionary distance to other non-mammalian vertebrates (Furutani-Seiki and Wittbrodt, 2004; Takeda and Shimada, 2010), which has an extensive collection of epigenome data on reprogramming during early development (Nakamura et al, 2014, 2021; Fukushima et al, 2023). We demonstrate here that it is not ZGA but regulation of cell cycle length that mediates both the erasure and re-accumulation of H3K9me3 during early development in medaka. We also show that this holds true for zebrafish and *Xenopus* embryos. Taken together with the previous studies in *C. elegans* and *Drosophila* (Mutlu et al, 2018, 2019; Seller et al, 2019), our results suggest that cell cycle length regulation commonly and mainly causes global erasure and re-establishment of heterochromatin in rapid-cleavage species, and provide a new evolutionary perspective on the dynamics of epigenetic reprogramming in animal development.

## Results

### Passive erasure of heterochromatin during early cleavage stages in medaka

Previously, we revealed that H3K9me3 is observed in medaka embryos at the one-cell stage, but is almost undetectable at the 16-cell stage, except in telomeric regions (Fukushima et al, 2023). To examine how H3K9me3 disappears between the two stages, we performed immunofluorescence staining of H3K9me3 at several developmental points between the two stages (Fig. 1A). We found that H3K9me3 was gradually decreased from the one-cell stage and became almost undetectable by the 8-cell stage, suggesting that H3K9me3 is globally erased by the 8-cell stage (Fig. 1B,C). The dynamics of H3K9me3 erasure is rather slow, compared to the rapid erasure of active mark H3K4me3 which occurs within the one-cell stage (Fig. EV1A,B), suggesting that active erasure is less likely to reduce H3K9me3 levels during the early cleavage stage. Instead, we reasoned that the passive dilution of H3K9me3 during every DNA replication accounts for the early erasure dynamics of H3K9me3, since medaka blastomeres rapidly divide every 30 min until the late morula stage (Iwamatsu, 2004).

Thus, we experimentally assessed the dilution hypothesis by prolonging cell cycles. If any active demethylation (e.g., by histone

demethylase) functions after fertilization, the prolonged cell cycle should result in a greater reduction of H3K9me3 levels when compared to the control at the same developmental stage. Conversely, if the passive dilution mainly works, H3K9me3 levels should not be affected in embryos with prolonged cell cycles at the same developmental stage. To slow down the cell cycle, we overexpressed Chk1 in cleavage-stage embryos, which inhibits the formation of replication origins (Kappas et al, 2000; Chan et al, 2019; Collart et al, 2017, 2013). In *chk1*-mRNA-injected embryos, the timing of early cleavages was delayed, suggesting the prolonged cell cycle (Fig. 1D). In these injected embryos, the level of H3K9me3 detected by immunofluorescence staining was not reduced, while H3K9me3 was almost completely erased in 8-cell stage control embryos, even though the same absolute time after fertilization had elapsed in both embryos (Fig. 1D–F, Control 2.1 hpf 8-cell vs Chk1 2.1 hpf 2-cell). Furthermore, prolonged cell cycles did not decrease the H3K9me3 level in injected embryos either, but slightly increased the H3K9me3 level, compared to that in 2-cell stage control embryos (Fig. 1D–F, Control 1.2 hpf 2-cell vs Chk1 2.1 hpf 2-cell, see Discussion). Moreover, the prolonged cell cycle did not disturb the rapid erasure of H3K4me3 (Fig. EV1C), suggesting that active erasure machinery, if any, was less likely to be affected by Chk1 overexpression. Taken together, these data suggest that the erasure of H3K9me3 in the early cleavage stages is mainly caused by DNA-replication-dependent dilution in medaka, and that H3K9me3 methyltransferases are active in de novo H3K9me3 deposition even during the erasure period (i.e., from the one-cell stage to the 8-cell stage) (discussed later).

### ZGA is dispensable for heterochromatin establishment during the MBT in medaka

In non-mammalian vertebrates, both ZGA and cell cycle slowing take place at the mid-blastula stage, which are collectively called 'mid-blastula transition' (MBT) (Tadros and Lipshitz, 2009; Vastenhouw et al, 2019). After the global decrease in H3K9me3 at early cleavage stages, medaka embryos begin to re-accumulate H3K9me3 again during the MBT (Fig. 2A) (Fukushima et al, 2023). Thus, we hypothesized that ZGA and/or the slowing of cell cycle triggers re-accumulation of H3K9me3 during the MBT. First, we investigated the role of ZGA in this process, as previously done in zebrafish (Laue et al, 2019). To block ZGA, we injected the RNA polymerase II inhibitor α-amanitin into medaka embryos (Nakamura et al, 2021) (Fig. EV2A,B). As expected, α-amanitin injection resulted in developmental arrest and injected embryos did not undergo gastrulation (Fig. 2B). The number and morphology of cells were comparable between control and α-amanitin-injected embryos until the late-blastula stage (just before the onset of gastrulation) (Figs. 2B and EV2C), indicating that the developmental arrest occurred at the stage of gastrulation. To avoid the effect of developmental arrest, we first compared H3K9me3 levels in control and α-amanitin-injected embryos at the late-blastula stage (the stage just before the developmental arrest becomes evident). As a result, both immunofluorescence staining and quantitative western blot confirmed that the H3K9me3 level was comparable between control and α-amanitin-injected embryos (Figs. 2C–F and EV2F,G). The same result was also obtained when compared at the pre-early gastrula stage (beginning of gastrulation) (Fig. EV2D,E). We thus conclude that ZGA, or active transcription

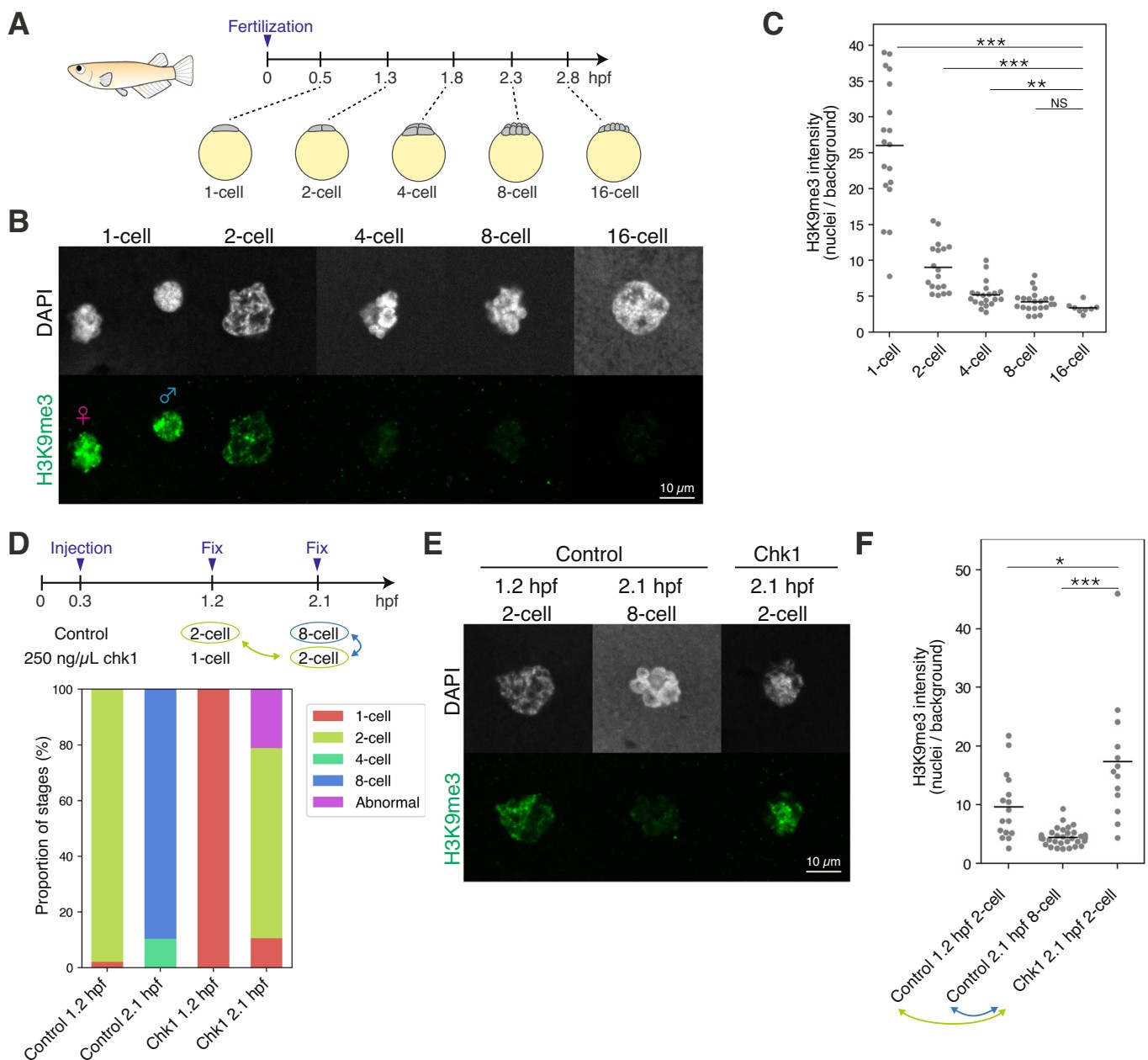

**Figure 1. Passive erasure of heterochromatin during early cleavage stages in medaka.**

(A) Development of medaka embryo during early cleavage stages. (hpf: hours post fertilization). (B) Immunofluorescence staining of H3K9me3 during early cleavage stages. (C) Quantification of (B). Each dot indicates the average intensity of 1–2 cells in a single broad field slice image of single embryo. Two-sided Wilcoxon rank-sum test. Bars indicate the means. n = 18, 18, 20, 22, 8 embryos for the 1, 2, 4, 8, and 16-cell stage, respectively. Data were pooled from three independent experiments. (D) Schematic summarizing the chk1 experiment (top) and the proportion of stages in the chk1 experiment in medaka (bottom). Stages highlighted in green and blue were compared in (E) and (F). (E) Immunofluorescence staining of H3K9me3 in the chk1 experiment. (F) Quantification of (E). Each dot indicates the average intensity of 1–2 cells in a single broad field slice image of single embryo. Two-sided Wilcoxon rank-sum test. Bars indicate the means. n = 16, 34, 13 embryos for the control 1.2 hpf, control 2.1 hpf and chk1 2.1 hpf, respectively. Data were pooled from three independent experiments. *p < 0.05, **p < 0.01, ***p < 0.001, NS: not significant. Source data are available online for this figure.

from the genome, is dispensable for re-accumulation of H3K9me3 during the MBT in medaka.

In the previous studies, in addition to H3K9me3 accumulation, DNA-dense domains in nuclei detected after ZGA was used to indicate the onset of heterochromatin formation (Mutlu et al, 2019; Laue et al, 2019). We also observed the formation of DAPI-dense

domains from the late blastula stage (i.e., after the MBT) in medaka embryos (Fig. 2G,H), which is ZGA-dependent as in zebrafish (Laue et al, 2019) (Fig. EV2H,I). However, artificial depletion of H3K9me3 (Fukushima et al, 2023) revealed that the formation of DNA-dense domains occurred irrespective of the presence or absence of H3K9me3 (Fig. 2I,J). This inconsistency

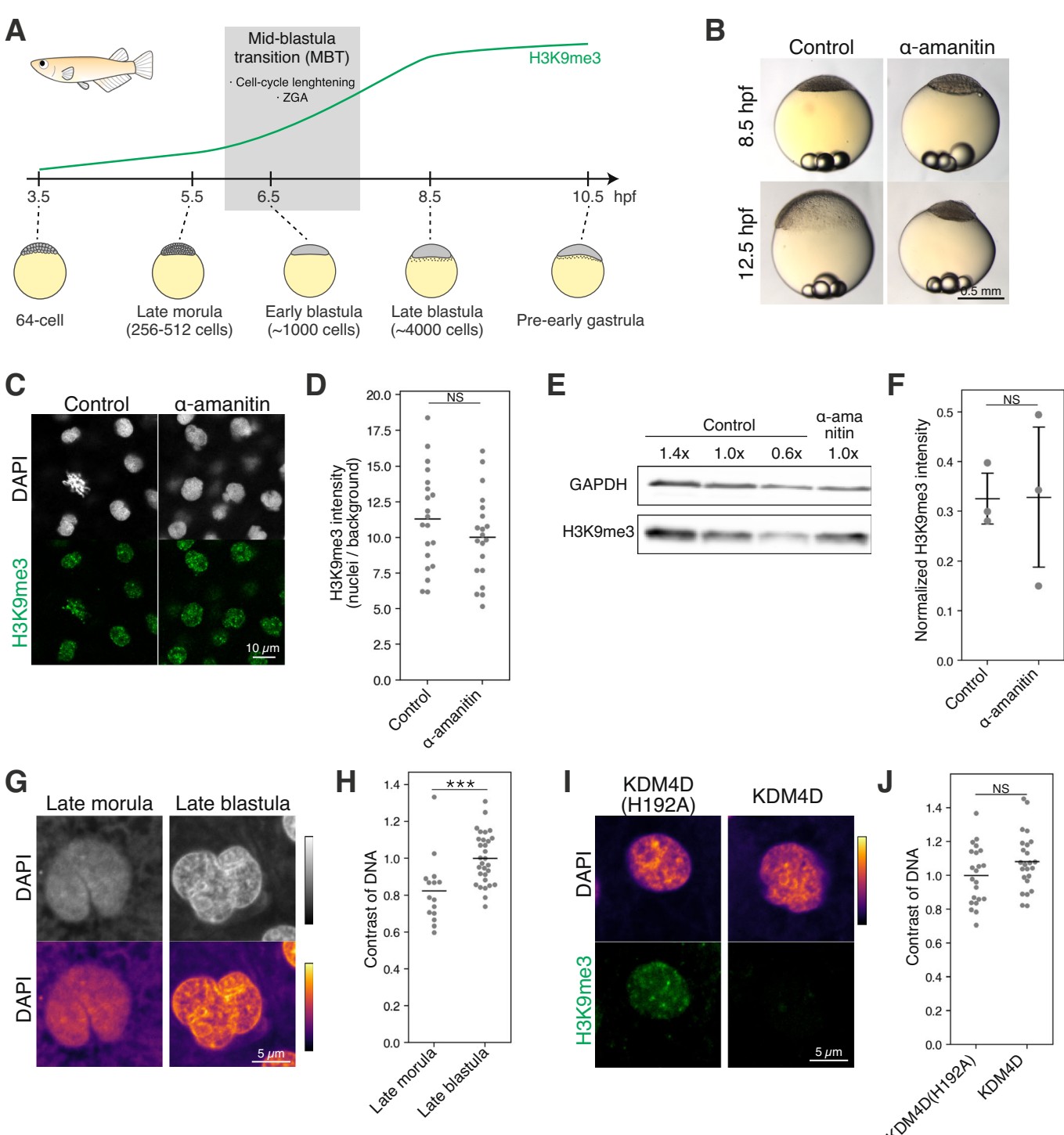

between presence of DNA-dense regions and presence of H3K9me3 is consistent with a recent study in zebrafish demonstrated that the formation of the DNA-dense domain after the MBT is dependent on micro-phase separation via zygotic transcription (Hilbert et al, 2021). Therefore, we concluded that it is not suitable to use the presence or absence of DNA-dense regions alone as an indicator of early-stage heterochromatin formation.

## Cell cycle slowing regulates heterochromatin establishment during MBT in medaka

Since ZGA is not required for H3K9me3 re-accumulation during the MBT, we next assessed if cell cycle slowing regulates this process. Except for earliest cleavages (~30 min/round) (Fig. 1A) (Iwamatsu, 2004), the cell cycle length during and after the MBT was not precisely determined in medaka. We first counted the cell

◄ **Figure 2. ZGA is dispensable for heterochromatin establishment during the MBT in medaka.**

(A) Development of medaka embryos before and after the MBT. (B) Phenotype of α-amanitin-injected medaka embryos. (C) Immunofluorescence staining of H3K9me3 at the late blastula stage (8.5 hpf) in the α-amanitin injection experiment. (D) Quantification of (C). Each dot indicates the average intensity of ~40 cells in a single broad field slice image of single embryo. Two-sided unpaired Student's t-test. Bars indicate the means. $n = 20$ embryos. Data were pooled from two independent experiments. (E) Western blot of H3K9me3 and GAPDH using control and α-amanitin-injected embryos at the late blastula stage (8.5 hpf). (F) Quantification of (E). H3K9me3 signal intensity was normalized by GAPDH signal intensity. Two-sided unpaired Student's t-test. Error bars indicate the mean ± s.d. $n = 3$ biological replicates. (G) DAPI-staining at the late morula and late blastula stages. Colormaps are shown at the bottom to better illustrate the appearance of DNA-dense regions at the late blastula stage. (H) Quantification of DNA contrast in (G). Each dot indicates the DNA contrast of a single nucleus. ~6 embryos were analyzed. Two-sided Wilcoxon rank-sum test. Bars indicate the means. $n = 15$ and 30 nuclei for the late morula and late blastula, respectively. Data were pooled from two independent experiments. (I) DAPI and immunofluorescence staining of embryos injected with human KDM4D, a demethylase of H3K9me3, or its catalytically inactive mutant KDM4D(H192A) at the late blastula stage. The pattern of DNA-dense domains was comparable irrespective of presence or absence of H3K9me3. (J) Quantification of DNA contrast in (I). Each dot indicates the DNA contrast of a single nucleus. Five embryos were analyzed. Two-sided unpaired Student's t-test. Bars indicate the means. $n = 22$ and 24 nuclei for the KDM4D(H192A) and KDM4D, respectively. Data were pooled from two independent experiments. ***$p < 0.001$, NS: not significant. Source data are available online for this figure.

number per embryo from 3.5 hpf (the 64-cell stage) to 10.5 hpf (the pre-early gastrula stage) (Fig. 2A) and found that the cell cycle was initially about 30 min/round, gradually prolonged from the 64-cell stage onward, followed by a dramatic slowing from 7.5 hpf, and finally reaching to ~1.7 h/round towards the late blastula stage (Fig. EV3A–C). Thus, there is an apparent correlation between the onset of H3K9me3 re-accumulation and cell cycle slowing in medaka embryos.

To test the causal relationship between the above two events, we prolonged the cell cycle by overexpressing Chk1 at a milder concentration than in the above experiment during the early cleavage stages in Fig. 1 (Kappas et al, 2000; Chan et al, 2019; Collart et al, 2017, 2013) and compared H3K9me3 levels by immunofluorescence staining. The developmental delay of *chk1*-injected embryos was evident from the 64-cell stage and onward (Fig. 3A), and did not affect the erasure of H3K9me3 by the 8-cell stage (Fig. EV3D,E), suggesting that the cell cycle was artificially prolonged after the erasure of H3K9me3 (Fig. 1A–C). In these *chk1*-injected embryos, we observed an increase in H3K9me3 levels at the morula stage (Fig. 3B,C, control 5.5 hpf late morula vs Chk1 8.5 hpf late morula). As an alternative approach, we extended cell cycle length by incubating embryos with the translation inhibitor, cycloheximide (CHX), from the 8-cell stage (i.e., after the erasure of H3K9me3 shown in Fig. 1A–C) (Chan et al, 2019). This treatment immediately slowed development, and again increased H3K9me3 levels at the 16-cell stage (Fig. 3D–F, DMSO 2.8 hpf 16-cell vs CHX 3.5 hpf 16-cell). Since H3K9me3 is normally erased by the 8-cell stage (Fig. 1A,B), the observed increase in H3K9me3 in both Chk1 and CHX experiments must not be caused by incomplete erasure of H3K9me3, but by precocious re-accumulation of H3K9me3. These experiments demonstrated that the slowing of the cell cycle is sufficient to induce H3K9me3 re-accumulation at the early stages.

The above results did not support the ZGA-dependent re-accumulation of H3K9me3, but the possibility still remained that H3K9me3 deposition is dependent on a "timer", or absolute time elapsed after fertilization. To further test the necessity of cell cycle slowing for H3K9me3 re-accumulation, we accelerated the cell cycle by overexpressing H3-tail, which competitively inhibits Chk1-dependent cell cycle slowing during the MBT (Shindo and Amodeo, 2021). Indeed H3-tail injection accelerated embryonic development at the cleavage stages (Fig. EV3F), and the number of cells in H3-tail-injected embryos was higher than that in control at the late

blastula stage (Fig. 3G), suggesting accelerated cell cycles. Under these conditions, H3K9me3 levels in H3-tail-injected embryos were reduced compared to the control at the late blastula stage, although the same absolute time had elapsed after fertilization (Fig. 3H,I). Thus, these data suggest that the slowing of the cell cycle, but not the elapsed absolute time, is critical for the sufficient re-accumulation of H3K9me3 upon the MBT.

The above cell cycle manipulations could induce replication stress or DNA damage which may affect the level of H3K9me3 accumulation. Indeed, it was reported that H3K9me3 depositions were induced by the DNA-repair pathway (Ayrapetov et al, 2014). We thus quantified the DNA damage marker, γH2AX, in the cell-cycle manipulation experiments. However, we did not find any statistically significant accumulation of γH2AX in the cell-cycle manipulated embryos (Fig. 3B,C,E,F). We also observed that DNA damage, induced by ultraviolet (UV) exposure of late-blastula embryos, increased γH2AX levels but did not H3K9me3 levels (Fig. EV3G,H), suggesting that DNA repair-mediated H3K9me3 deposition is less likely to be active in early medaka embryos.

The cell cycle manipulations could affect the time window of ZGA as previously reported (Chan et al, 2019; Chen and Good, 2022; Strong et al, 2020). Therefore, we compared the expression level of some zygotic genes, *sox2*, *zic2a*, and *tbx16*, and found that Chk1 overexpression caused precocious expression of zygotic genes from the late morula stage, although the extent is relatively mild, while CHX treatment did not at the 16-cell stage (Fig. EV3I). These data rule out the possibility that H3K9me3 accumulation after cell cycle slowing was caused by premature initiation of ZGA. Taken together, we conclude that the slowing of the cell cycle causes re-accumulation of H3K9me3 upon the MBT in medaka.

## Setdb1 accumulates in nuclei during the MBT in medaka

To elucidate the molecular mechanism underlying the onset of H3K9me3 deposition during the MBT, we focused on the evolutionary conserved H3K9me3 methyltransferases Suv39h1/2 and Setdb1, which are known to be essential for re-accumulation of H3K9me3 in mouse, and *Drosophila* and *C. elegans*, respectively (Mutlu et al, 2018, 2019; Seller et al, 2019; Burton et al, 2020).

In medaka, Setdb1 and Suv39h1 are encoded by *setdb1b* and *suv39h1b*, respectively. As we previously showed (Fukushima et al, 2023), the expression levels of *setdb1b*, *suv39h1b* and H3K9me3-related histone demethylases are almost comparable before and

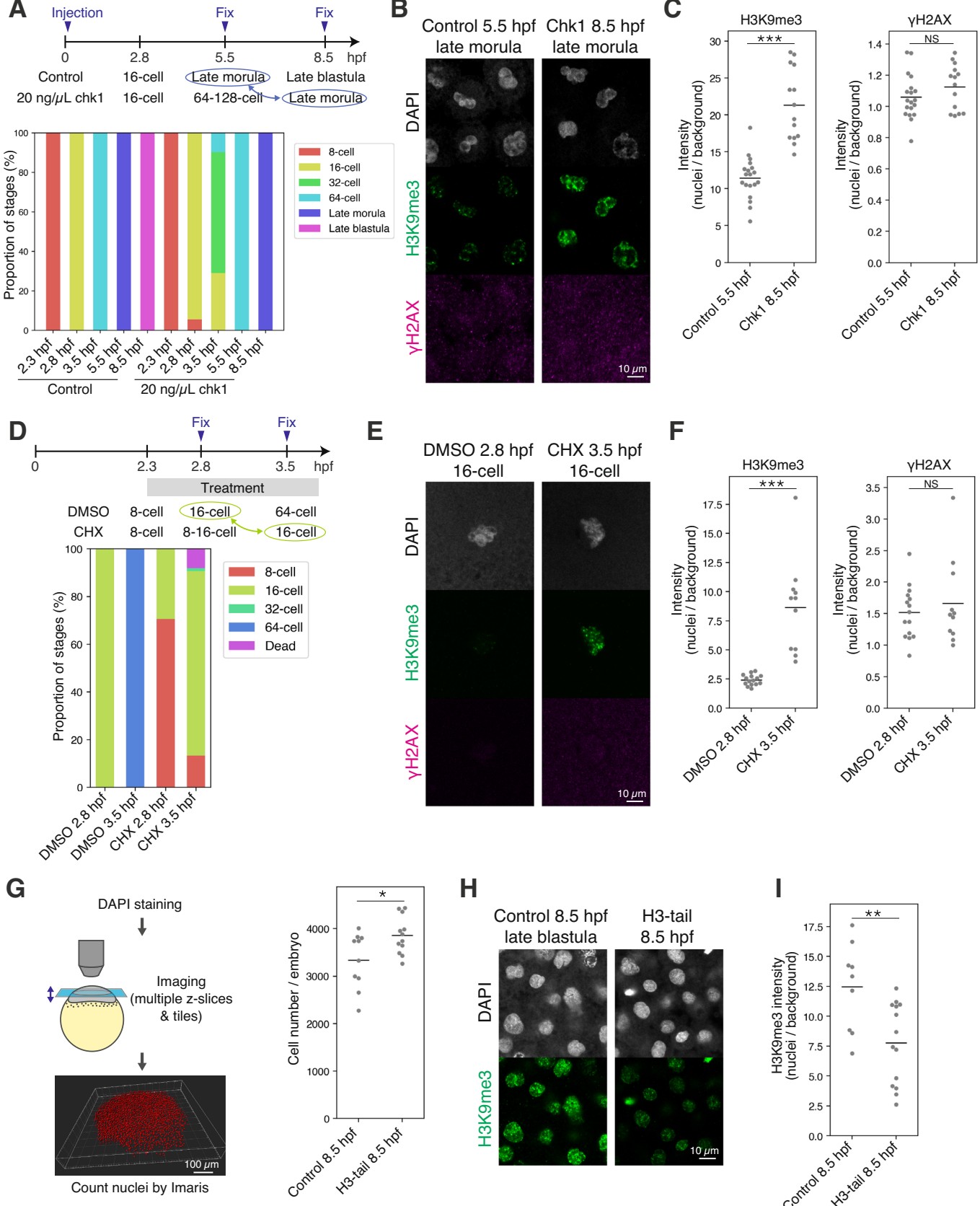

◀ **Figure 3. Cell cycle slowing regulates heterochromatin establishment during the MBT in medaka.**

(A) Schematic summarizing the chk1 experiment (top) and the proportion of stages in the chk1 experiment in medaka (bottom). Stages highlighted in blue were compared in (B) and (C). (B) Immunofluorescence staining of H3K9me3 and γH2AX at the late morula stage in the chk1 injection experiment. (C) Quantification of (B). Each dot indicates the average of 30–40 cells in a single broad field slice image of single embryo. Two-sided Welch's t-test and two-sided unpaired Student's t-test were performed for H3K9me3 and γH2AX, respectively. Bars indicate the means. $n = 19$ and 14 embryos for the control 5.5 hpf and chk1 8.5 hpf, respectively. Data were pooled from two independent experiments. (D) Schematic summarizing CHX treatment (top) and proportion of stages in CHX treatment in medaka (bottom). Stages highlighted in green were compared in (E) and (F). (E) Immunofluorescence staining of H3K9me3 and γH2AX in CHX treatment at the 16-cell stage. (F) Quantification of (E). Each dot indicates the average of ~5 cells in a single broad field slice image of single embryo. Two-sided Welch's t-test and two-sided Wilcoxon rank-sum test were performed for H3K9me3 and γH2AX, respectively. Bars indicate the means. $n = 15$ and 11 embryos for the DMSO 2.8 hpf and CHX 3.5 hpf, respectively. Data were pooled from three independent experiments. (G) Schematic showing counting of nuclei in an embryo (left) and the number of cells per embryo in the H3-tail injection experiment (8.5 hpf) (right). Two-sided unpaired Student's t-test. Bars indicate the means. $n = 10$ and 12 embryos for the control 8.5 hpf and H3-tail 8.5 hpf, respectively. Data were pooled from two independent experiments. (H) Immunofluorescence staining of H3K9me3 in H3-tail injection (8.5 hpf). (I) Quantification of (H). Each dot indicates the average intensity of ~140 cells in a single broad field slice image of single embryo. Two-sided unpaired Student's t-test. Bars indicate the means. $n = 9$ and 14 embryos for the control 8.5 hpf and H3-tail 8.5 hpf, respectively. Data were pooled from two independent experiments. $*p < 0.05$, $**p < 0.01$, $***p < 0.001$, NS: not significant. Source data are available online for this figure.

after the MBT (Fig. EV4A). Furthermore, we found that H3K9me3 re-accumulation proceeded even under conditions of translational inhibition by CHX treatment (Fig. 3D–F). Thus, neither transcriptional nor translational regulation of *setdb1b* and *suv39h1b* could account for the cell cycle-dependent H3K9me3 deposition during the MBT in medaka.

As another possibility, we focused on post-translational regulation of the histone methyltransferases. Indeed, Setdb1 has both a nuclear export signal (NES) and a nuclear localization signal (NLS) at its N-terminus (Fig. EV4B), and the nuclear and cytoplasmic localization of Setdb1 and Eggless (Setdb1 homolog in *Drosophila*) is known to be tightly regulated in mice and *Drosophila*, respectively (Osumi et al, 2019; Tachibana et al, 2015; Cho et al, 2013; Tsusaka et al, 2019). Importantly, nuclear accumulation of Eggless and MET-2 (Setdb1 homolog in *C. elegans*) has been implicated in re-accumulation of H3K9 methylations in early development of *Drosophila* and *C. elegans*, respectively (Mutlu et al, 2018, 2019; Seller et al, 2019). We thus reasoned that nuclear accumulation of maternally provided Setdb1 is associated with the onset of H3K9me3 accumulation during the MBT in medaka, and performed imaging of Setdb1 localization by immunofluorescence staining. After having validated the antibody specificity (Fig. EV4B,C), we found that before the MBT, Setdb1 was excluded from the nuclei and mainly localized in the cytoplasm, whereas after the MBT it was localized in both nuclei and cytoplasm (Fig. 4A–C). As shown above, the cell cycle length after the MBT becomes much longer than that before the MBT: the late morula stage (5.5 hpf), the late blastula stage (8.5 hpf) and the pre-early gastrula stage (10.5 hpf) is ~0.6, 1.7, and 2.6 h/round (Fig. EV3A–C). This correlation led us to test the causality between cell cycle slowing and the onset of Setdb1 nuclear localization, and we found that cell cycle extension by *chk1* overexpression promoted Setdb1 nuclear localization (Fig. 4D). Contrary to Setdb1, Suv39h1 does not have NES (Padeken et al, 2022). Consistently, overexpressed FLAG-tagged Setdb1 exclusively localized to the cytoplasm like endogenous one, whereas overexpressed FLAG-tagged Suv39h1 localized to the nuclei at the late morula stage (before the MBT) (Fig. EV4D), suggesting that Suv39h1 contributes minimally to the onset of de novo H3K9me3 deposition. These results suggest that, as in *C. elegans* and *Drosophila*, the slowing of the cell cycle induces nuclear accumulation of Setdb1, and thereby leads to re-accumulation of H3K9me3 at the MBT in medaka (Fig. 4E).

## Heterochromatin establishment during the MBT is cell cycle length-dependent in zebrafish

Contrary to our data in medaka, the previous study with zebrafish showed by western blot that H3K9me3 re-accumulation during the MBT depends on ZGA (Laue et al, 2019). To address this discrepancy, we first reproduced the previous zebrafish experiment. We first confirmed in zebrafish that H3K9me3 accumulated during the MBT (Fig. 5A–C) and then attempted to block ZGA by α-amanitin injection (Laue et al, 2019; Chan et al, 2019; Zhang et al, 2018; Pálfy et al, 2020). This resulted in a reduction in the number of cells per embryo at the dome stage (the onset of epiboly) (Fig. EV5A,B), and injected embryos did not undergo gastrulation (Fig. 5D), indicating that α-amanitin injection leads to developmental arrest at the stage of gastrulation. Under these experimental conditions, we compared H3K9me3 levels between control and α-amanitin-injected embryos at the sphere stage (just before gastrulation) by immunostaining, but found no statistically significant changes (Fig. 5E,F). We next compared H3K9me3 levels at the dome stage, which is the stage used in the previous study (Laue et al, 2019), but again found no differences (Fig. EV5C,D). Our imaging analysis quantified the average signal in nuclei (intensity in nuclei/area of nuclei, see the Methods for detail) and was not affected by the number of cells per embryo, whereas the previous study (Laue et al, 2019) quantified the levels of H3K9me3 by western blot, which is potentially sensitive to the cell number fluctuations. Indeed, our quantitative western blot, in which samples from the same number of embryos at the dome stage were loaded in each lane, showed a reduction in H3K9me3 after α-amanitin injection (Fig. EV5E–G) (the similar result as in the previous zebrafish experiment). However, H3K9me3 levels became comparable when normalized by the cell number per embryo based on our data (Fig. EV5B,G). These data indicate that the previous finding of a reduction in H3K9me3 after α-amanitin injection was due to developmental arrest and resulting difference in the number of cells per embryos between control and α-amanitin-injected embryos at the dome stage, but not due to the inhibition of H3K9me3 re-accumulation. Indeed, in spite of cell cycle slowing at the MBT, the number of cells per embryo continuously increase after ZGA in medaka (Fig. EV3A) and zebrafish (Joseph et al, 2017; Keller et al, 2008). Taken together, we conclude that the re-accumulation of H3K9me3 during the MBT is ZGA-independent in zebrafish.

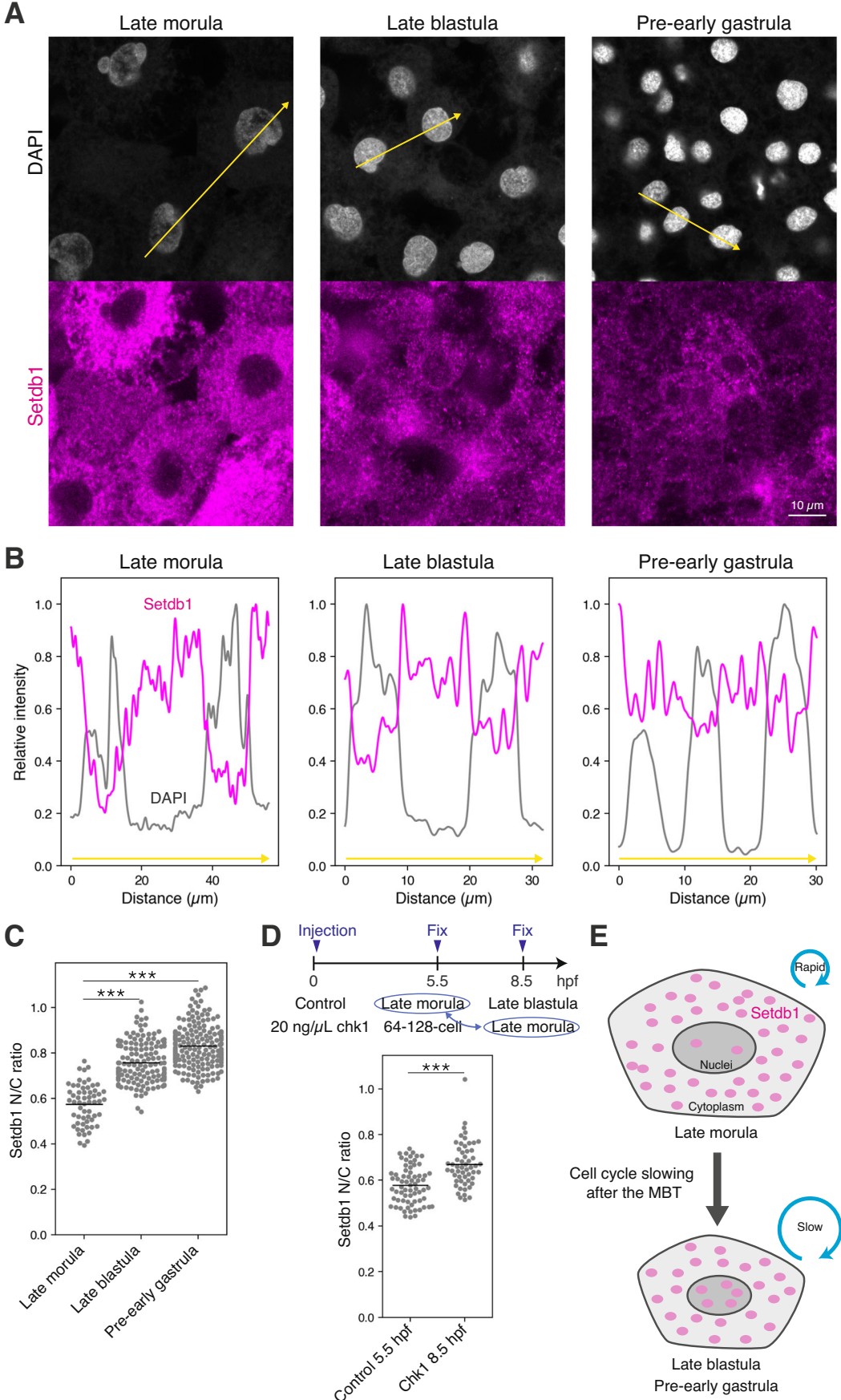

**Figure 4.  Setdb1 accumulates to nuclei upon the MBT in medaka.**

(A) Immunofluorescence staining of Setdb1 in medaka embryos before (late morula) and after the MBT (late blastula and pre-early gastrula). Signal intensities along the yellow arrows were quantified in (B). (B) Quantification of signal intensity of DAPI and Setdb1 along the yellow arrows in (A). (C) Quantification of nuclear/cytoplasmic ratio (N/C ratio) of Setdb1 in medaka embryos before (late morula) and after the MBT (late blastula and pre-early gastrula). Each dot indicates the N/C ratio of a single cell. 10, 8, and 10 embryos at the late morula, late blastula, and pre-early gastrula, respectively, were analyzed. Two-sided Wilcoxon rank-sum test. Bars indicate the means. $n = 59$, 142, and 193 cells for the late morula, late blastula, and the pre-early gastrula stage, respectively. Data were pooled from three independent experiments. (D) Schematic showing chk1 overexpression experiment (top) and quantification of N/C ratio of Setdb1 in control and *chk1*-injected embryos (bottom) at the late morula stage. Each dot indicates the N/C ratio of a single cell. Eleven embryos were analyzed for each condition. Two-sided Wilcoxon rank-sum test. Bars indicate the means. $n = 69$ and 55 cells for the control 5.5 hpf and chk1 8.5 hpf, respectively. Data were pooled from two independent experiments. (E) Schematic representation of the model of Setdb1 accumulation induced by cell cycle slowing during the MBT. ***$p < 0.001$. Source data are available online for this figure.

Next, we also tested the cell cycle length dependency of H3K9me3 re-accmulation in zebrafish, and obtained the results similar to those in medaka; CHX treatment from the 8-cell stage prolonged the length of the cell cycle in zebrafish embryos (Fig. 5G), and increased H3K9me3 levels at the 16-cell stage without apparent DNA damage (Fig. 5H,I). This suggests that the prolonged cell cycle length is sufficient to trigger H3K9me3 deposition in zebrafish. Taken together, the cell cycle length-dependent and ZGA-independent re-accumulation of H3K9me3 is a conserved feature at least among teleosts.

## Heterochromatin establishment during the MBT is cell cycle length-dependent in *Xenopus laevis*

Finally, we extended our analysis to amphibian embryos, *Xenopus laevis*, using immunostaining. As in medaka and zebrafish, H3K9me3 levels gradually increase during the MBT in *Xenopus* (Fig. 6A–C). As previously reported (Sudou et al, 2016; Chen et al, 2019), blocking ZGA by α-amanitin injection resulted in developmental arrest, and injected embryos did not undergo gastrulation (Fig. 6D). We tested the requirement of ZGA for H3K9me3 accumulation, but we did not find statistically significant differences in H3K9me3 levels between control and α-amanitin-injected embryos at the stage 10 (the onset of gastrulation) (Fig. 6B,C). We conclude that ZGA is dispensable for H3K9me3 re-accumulation during the MBT also in *Xenopus laevis*.

Next, we assessed the cell cycle length dependency by experimentally manipulating the cell cycle length. For this purpose, we first applied *chk1* overexpression in *Xenopus* embryos. The *chk1*-mRNA-injection increased the cell size, suggesting that Chk1 overexpression prolonged cell cycles (Fig. 6E). However, the size varies, and animal pole tends to be more sensitive to Chk1 overexpression than vegetable pole (Fig. 6E). Therefore, for comparison, we selected cells in *chk1*-injected embryos (9 hpf) whose size is almost comparable to that in normal embryos at 7 hpf and found that H3K9me3 levels were higher in *chk1*-injected embryos than that in 7 hpf-normal embryos (Fig. 6F,G), indicating that cell cycle slowing caused precocious accumulation of H3K9me3. The same result was obtained by CHX treatment from the 64-cell stage; prolonged cell cycles (Fig. EV5H) caused precocious accumulation of H3K9me3 (Fig. EV5H–J). Collectively, the mechanism of H3K9me3 re-accumulation during the MBT in *Xenopus* is likely to be similar to that in teleosts, i.e., ZGA-independent and cell cycle length-dependent.

## Discussion

The proper reboot of the epigenetic memory after fertilization is an essential process for early embryogenesis, but its molecular mechanisms were not fully understood in vertebrates. In this study, we have shown which factor mediates heterochromatin erasure and re-establishment during early development in medaka, zebrafish, and *Xenopus*. We first showed that the erasure of H3K9me3 was caused by DNA replication-dependent passive dilution. Second, we revealed that it is cell cycle slowing, not ZGA, that triggers the re-accmulation of H3K9me3 during the MBT. When rapid embryonic cell cycles slow down toward the MBT, nuclear localization of maternal Setdb1 and accumulation of H3K9me3 simultaneously became detectable. This led us to propose the model that Setdb1 accumulated in nuclei during prolonged cell cycles promotes the deposition of H3K9me3. In other words, rapid cell cycles limit nuclear accumulation of Setdb1, but prolonged cell cycle permits accumulation of Setdb1, leading to the onset of H3K9m3 deposition. Consistent with this model, experimental prolongation of the cell cycle increased H3K9me3 levels at the early cleavage stages, probably because the extension of cell cycle allowed the accumulation of Setdb1 at the early cleavage stages (Fig. 1D–F). The data obtained in zebrafish, medaka and *Xenopus* are all consistent with each other. Our study thus provides experimental evidence for the essential role of cell cycle length in both the erasure and re-establishment of heterochromatin during early development in non-mammalian vertebrates (Fig. 7).

We showed that ZGA is dispensable for the onset of H3K9me3 deposition in non-mammalian vertebrates. In contrast, it was previously reported in zebrafish that zygotic transcription of miR-430 and its mediated degradation of maternal factors are essential for this process (Laue et al, 2019). As described in Results, this discrepancy can be resolved by taking into account developmental arrest and the resulting reduction in cell number. However, we do not exclude the possibility that the degradation of maternal factors is also involved in the re-accumulation of H3K9me3. Furthermore, the mechanism of nuclear accumulation of Setdb1 upon cell cycle slowing is yet to be determined. However, the modest increase in N/C ratio upon cell cycle slowing (Fig. 4) prompts us to speculate that, unlike the active translocation of Setdb1 observed in cell lines (Osumi et al, 2019; Tachibana et al, 2015; Cho et al, 2013; Tsusaka et al, 2019), certain passive import machinery would be involved in at least in medaka embryos. Future studies are highly warranted.

Similar to non-mammalian vertebrates, invertebrate embryos such as *Drosophila* and *C. elegans* undergo rapid cleavages in early development (Tadros and Lipshitz, 2009; Vastenhouw et al, 2019) and re-install H3K9 methylation after 5–10 rounds of cleavages (Mutlu et al, 2018, 2019; Seller et al, 2019). In these species, cell cycle slowing also times heterochromatin formation, which is mediated by translocation of Setdb1 homolog from the cytoplasm to nuclei (Mutlu et al, 2018, 2019; Seller et al, 2019). Therefore, the

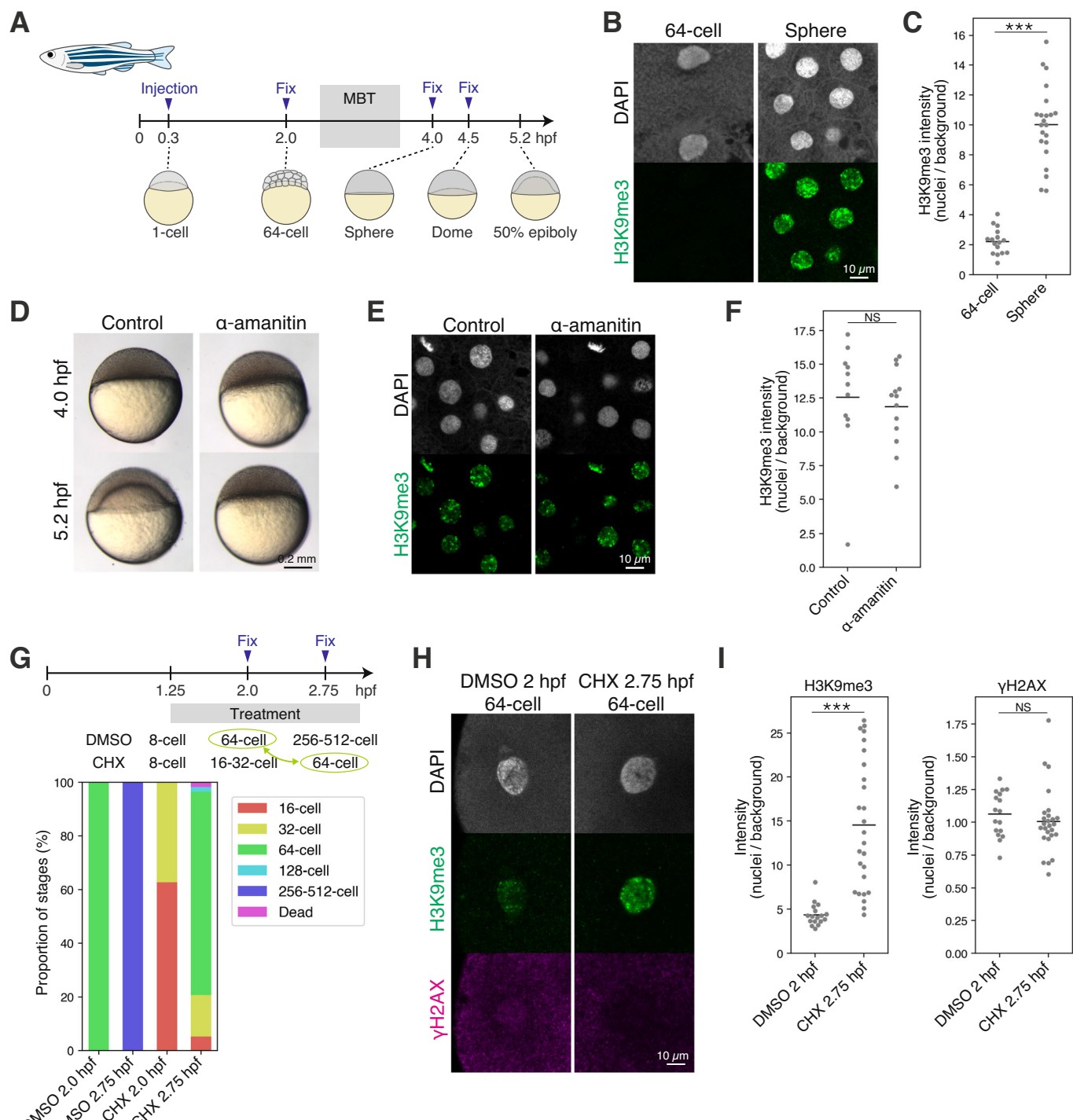

molecular mechanism of H3K9me3 re-accumulation during epigenetic reprogramming is broadly conserved in rapid-cleavage species. During rapid cleavage, transcription machinery is robustly inactivated before ZGA, due to the absence of "activators" (Jukam et al, 2017), lack of active histone modifications (Chan et al, 2019), excessive nucleosome (Joseph et al, 2017; Amodeo et al, 2015; Wilky et al, 2019), insufficient nuclear import machinery (Shen et al, 2022) and so on. Under these conditions, heterochromatin-mediated gene silencing is not necessarily required. Furthermore,

active demethylation is also not required (Mutlu et al, 2019), as passive dilution is sufficient to erase H3K9me3 globally and almost completely by the onset of ZGA (after several rounds of cell division). The only exception reported so far is the retention of H3K9me3 at the telomeric region, which is essential for the maintenance of genomic stability (Fukushima et al, 2023). This specific retention can be interpreted in the context of the passive erasure, if we assume that the original level of H3K9me3 accumulation is high enough prior to reprogramming. Taken

**Figure 5. Heterochromatin establishment during the MBT is cell cycle length-dependent in zebrafish.**

(A) Development of zebrafish embryo before and after the MBT. (B) Immunofluorescence staining of H3K9me3 in zebrafish embryos before (64-cell) and after the MBT (Sphere). (C) Quantification of (B). Each dot indicates the average of ~5 and ~60 cells in a single broad field slice image of single embryo at the 64-cell stage and the sphere stage, respectively. Two-sided Welch's t-test. Bars indicate the means. $n = 16$ and 22 embryos for the 64-cell and sphere stage, respectively. Data were pooled from two independent experiments. (D) Phenotype of α-amanitin-injected zebrafish embryos. (E) Immunofluorescence staining of H3K9me3 in the α-amanitin injection experiment at the sphere stage. (F) Quantification of (C). Each dot indicates the average of ~60 cells in a single broad field slice image of single embryo. Two-sided Wilcoxon rank-sum test. Bars indicate the means. $n = 11$ and 13 embryos for the control and α-amanitin, respectively. Data were pooled from two independent experiments. (G) Schematic summarizing the CHX experiment (top) and the proportion of stages of CHX-treated zebrafish embryos (bottom). Stages highlighted in green were compared in (H) and (I). (H) Immunofluorescence staining of H3K9me3 and γH2AX in CHX treatment at the 64-cell stage. (I) Quantification of (H). Each dot indicates the average of ~2 cells in a single broad field slice image of single embryo. Two-sided Wilcoxon rank-sum test. Bars indicate the means. $n = 17$ and 26 embryos for the DMSO 2 hpf and CHX 2.75 hpf, respectively. Data were pooled from two independent experiments. ***$p < 0.001$, NS: not significant. Source data are available online for this figure.

together, the passive erasure, which is characterized by a uniform reduction of H3K9me3 levels along the entire genomic region, would be more cost-effective in rapid-cleavage species.

By contrast, in mammalian embryos, postfertilization cleavages are much slower, and ZGA is initiated in a few rounds of cleavage (Tadros and Lipshitz, 2009; Vastenhouw et al, 2019). In this case, the chance of passive erasure is limited, and instead, active demethylation comes into play during the cleavage stages (Sankar et al, 2020; Liu et al, 2018). This active erasure can work site-specifically and as a result, in mammals H3K9me3 undergo reprogramming in a non-uniform manner (Wang et al, 2018; Yu et al, 2022; Zhou et al, 2023). Mammals take advantage of this mechanism to ensure H3K9me3 retention in specific genomic regions where it is needed, such as silencing of LTRs (Wang et al, 2018) and maintenance of DNA methylation as CpG-rich loci during cleavage stages (Yang et al, 2022). In addition, de novo H3K9me3 deposition begins immediately after fertilization using Suv39h1/2 (Burton et al, 2020). Therefore, we propose a scenario of stepwise evolution of H3K9me3 reprogramming dynamics; the cell cycle-dependent H3K9me3 reprogramming observed in rapid-cleavage species evolved first, and mammals later adopted active demethylation to adjust slow cleavage and immediate ZGA.

## Methods

### Animal procedures

Medaka d-rR strain was used in this study. Fertilized embryos were raised according to standard protocols (Kinoshita et al, 2009) at 28 °C. Developmental stages were determined according to previously published guidelines (Iwamatsu 2004) (Figs. 1A, 2A and EV1A). 10 ng/μL α-amanitin (Nakamura et al, 2021), 200 ng/μL *H3-tail* mRNA, 250 or 20 ng/μL medaka's *chk1* mRNA, 131 ng/μL (300 nM) *FLAG-suv39h1b* mRNA, 358 ng/μL (300 nM) *FLAG-setdb1b* mRNA, 514 ng/μL human *KDM4D* mRNA, and 514 ng/μL human *KDM4D(H192A)* mRNA (Fukushima et al, 2023) were injected into one-cell stage embryos. Dechorionated embryos were incubated with 20 ng/μL CHX from the 8-cell stage.

Zebrafish RW strain was used in this study. Fertilized embryos were raised according to standard protocols (Kimmel et al, 1995) at 28.5 °C. Developmental stages were determined according to previously published guidelines (Kimmel et al, 1995) (Fig. 5A). One-cell stage embryos were injected with 200 pg α-amanitin (Laue et al, 2019; Chan et al, 2019; Zhang et al, 2018; Pálfy et al, 2020). Dechorionated embryos were incubated with 20 ng/μL CHX from the 8-cell stage.

Wild-type *X. laevis* was used in this study. In vitro fertilized embryos were raised at 23 °C. Developmental stages were determined according to previously published guidelines (Nieuwkoop and Faber, 1994) (Fig. 6A). At the two-cell stage, 100 pg α-amanitin (Sudou et al, 2016; Chen et al, 2019) and 200 pg medaka *chk1* mRNA were injected into each blastomere. Embryos were incubated with 10 ng/μL CHX from the 64-cell stage.

All experimental procedures and animal cares were performed under the approval of the animal ethics committee of the University of Tokyo (Approval No. 20-2).

### Constructions and in vitro transcription

Medaka *chk1*, *setdb1b*, and *suv39h1b* sequences were amplified by PCR from the medaka cDNA library, cloned into TOPO vector using TOPO TA Cloning Kit Dual Promoter (Invitrogen, 45-0640), and introduced to pCS2+ vector using NEBuilder HiFi DNA Assembly Master Mix (NEB, E2621). The H3-tail sequence (Shindo and Amodeo, 2021) was generated by PCR using human histone H3.1 sequence as a template and inserted directly into pCS2+ vector using NEBuilder HiFi DNA Assembly Master Mix. Templates for in vitro transcription were amplified by PCR, and mRNA was transcribed in vitro from the templates using HiScribe T7 ARCA mRNA Kit with Tailing (NEB, E2060) or mMESSAGE mMACHINE SP6 Transcription Kit (Thermo, AM1340). Previously generated plasmids (Fukushima et al, 2023) were used for in vitro transcription of human *KDM4D* and *KDM4D(H192A)* mRNA. All primer sequences used for construction are listed in Table EV1. The transcribed mRNA was purified using RNeasy mini kit (QIAGEN).

### Immunofluorescence staining and imaging

Immunofluorescence staining was performed according to the previous protocol (Fukushima et al, 2023) with minor modifications. Briefly, embryos fixed with 4% PFA/PBS were permeabilized with 0.5% Triton X-100/PBS for 30 min at room temperature, washed with PBS, incubated with blocking buffer (2% BSA, 1% DMSO, 0.2% Triton X-100, 1×PBS) for 1 h at room temperature, incubated with primary antibodies overnight at 4 °C, washed with PBSDT (1×PBS, 1% DMSO, 0.1% Triton X-100), incubated with blocking buffer for 1 h at room temperature, incubated with secondary antibodies and DAPI for 4 h at 4 °C, and washed with PBSDT. For immunofluorescence staining of endogenous Setdb1, after permeabilization, samples were incubated with 4 M HCl for 15 min at room temperature and incubated with 100 mM Tris-HCl for 20 min at room temperature for antigen retrieval. For

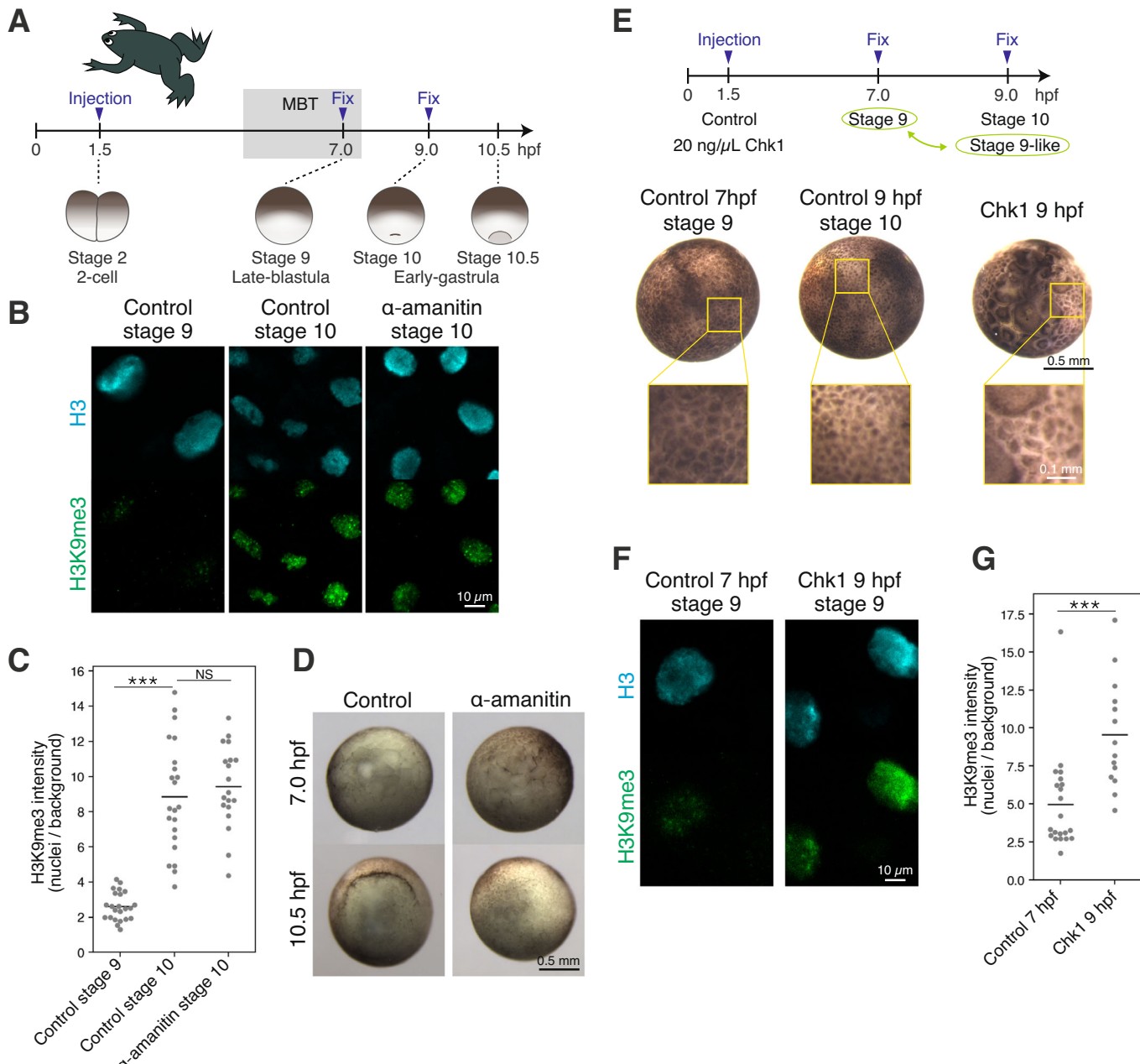

**Figure 6. Heterochromatin establishment during the MBT is cell cycle length-dependent in *X. laevis*.**

(A) Schematic illustration of *X. laevis* embryo development before and after the MBT. (B) Immunofluorescence staining of H3K9me3 in control (stage 9 and stage 10) and α-amanitin-injected (stage 10) *X. laevis* embryo. (C) Quantification of (B). Each dot indicates the average of ~20 cells in a single broad field slice image of single embryo. Two-sided Welch's t-test. Bars indicate the means. n = 23, 22 and 19 embryos for the control stage 9, control stage 10 and α-amanitin stage 10, respectively. Data were pooled from two independent experiments. (D) Phenotype of α-amanitin-injected *X. laevis* embryos. Unlike control embryos, α-amanitin-injected embryos failed to form dorsal lip at 10.5 hpf, indicating defects in gastrulation. (E) Schematic summarizing *chk1* injection (top) and animal view of *chk1*-injected *X. laevis* embryos (bottom). Stages highlighted in green were compared in (F) and (G). To compare developmental stages, cells at the animal poles are shown in magnified views (yellow squares). (F) Immunofluorescence staining of H3 and H3K9me3 in the *chk1* injection at the stage 9. (G) Quantification of (F). Each dot indicates the average of ~10 cells in a single broad field slice image of single embryo. Two-sided Wilcoxon rank-sum test. Bars indicate the means. n = 21 and 14 embryos for the control 7 hpf and chk1 9 hpf, respectively. Data were pooled from three independent experiments. ***p < 0.001, NS: not significant. Source data are available online for this figure.

immunofluorescence staining of medaka embryos, blastodiscs were manually separated from yolk and mounted on slide glasses with coverslips. For immunofluorescence staining of *X. laevis* embryos, whole embryos were mounted on slide glasses with coverslips. For

immunofluorescence staining of zebrafish embryos, samples were mounted in 1% low-melting point agarose/PBS. Imaging was performed with a Zeiss LSM710 or Leica SP8. Total cell count images are z-stacked and tiled images, and the other images are

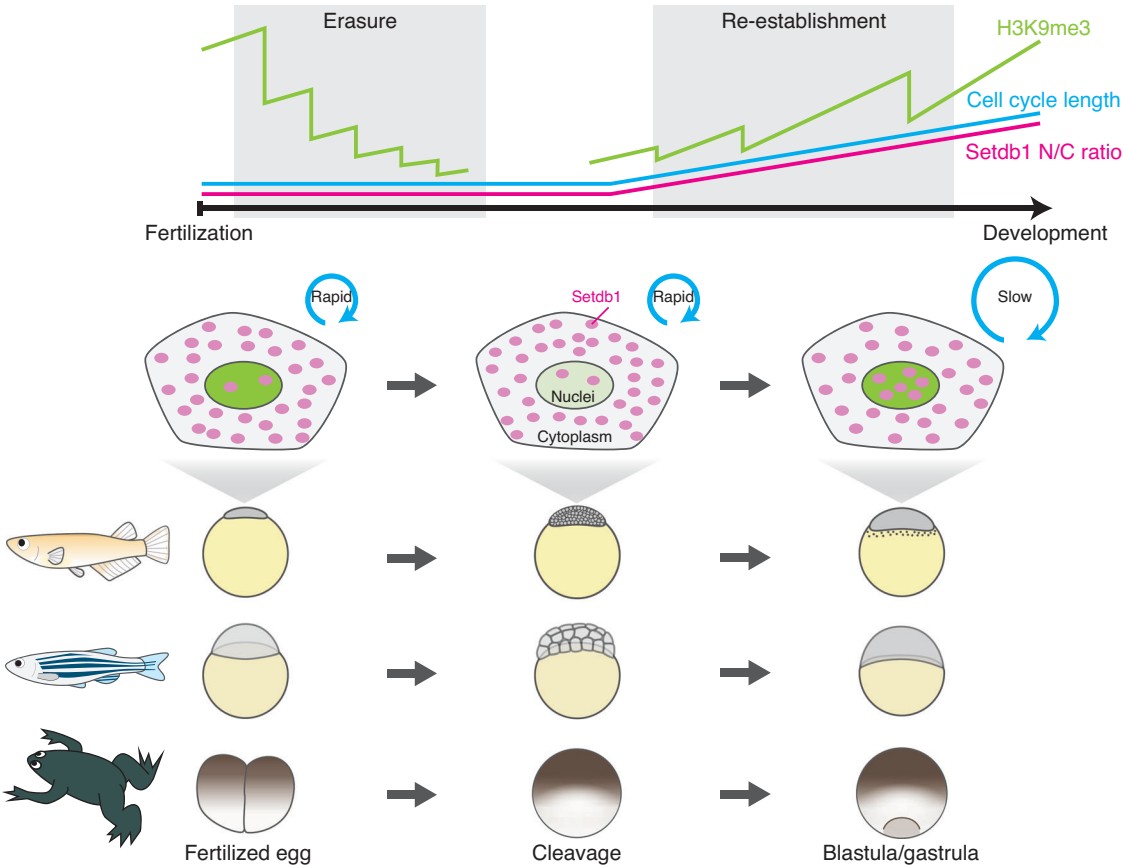

**Figure 7. Cell cycle length governs both erasure and re-establishment of heterochromatin during early development in non-mammalian vertebrates.**

Schematic summarizing the model of H3K9me3 reprogramming in non-mammalian vertebrates. Cell cycles in fertilized eggs and cleavage embryos are very rapid in non-mammalian vertebrates. This prevents Setdb1 (magenta dots) from accumulating in nuclei, resulting in DNA replication-dependent gradual erasure of H3K9me3 (green intensity in nuclei). However, cell cycles were prolonged from the MBT. Hereafter, Setdb1 can sufficiently accumulate in nuclei during the slowing of cell cycles, increasing H3K9me3 levels in blastula and gastrula embryos.

single-slice images. Antibodies used for immunofluorescence are listed in Table EV2.

## UV exposure

8.25 hpf medaka embryos were placed in clean bench and exposed UV for 10 min, and incubated for 5 min at 28 °C. The UV-exposed embryos were then fixed, and immunofluorescence staining was performed as described above.

## Quantification of imaging

Images were quantified using Fiji (Schindelin et al, 2012) according to a previous protocol (Fukushima et al, 2023) with minor modifications. For medaka and zebrafish samples, DAPI-dense regions were automatically or manually selected as nuclei, signal intensities of histone modifications in nuclei were measured, and the averages were normalized by signal intensities in manually selected background areas. Because the DAPI signal was relatively weak and noisy in *X. laevis* samples, H3-dense regions were automatically or manually selected as nuclei, and signals were measured as above. To measure the N/C ratio of Setdb1, DAPI-dense regions and surrounding regions

were manually selected, signal intensities and area sizes were measured, and the N/C ratio was calculated.

## Cell number estimation by qPCR

The same number of dechorionated embryos were homogenized in microtubes by gentile pipetting up and down in ice-cold PBS. After centrifugation, the supernatant was removed, and the pellet was stored in a freezer. The cell pellet was lysed with lysis buffer (50 mM Tris-HCl pH 8.0, 10 mM EDTA, 1% SDS) and sonicated briefly with Covaris (peak power: 105, duty factor: 4.0, cycles per burst: 200, duration: 180 s). After RNaseA treatment and ProK treatment, DNA was purified by phenol: chloroform: isoamyl alcohol method and ethanol purification. The relative amount of DNA was measured by qPCR using AriaMx (Agilent). Primers used in this study are listed in Table EV1.

## Total cell number counting

We counted the total number of cells per medaka embryo from images following the protocol of a previous study (Joseph et al, 2017). Briefly, tile-scan and z-stack DAPI images were acquired

using immunofluorescence staining samples, the figures were stitched using ZEN 2.3 SP1 (ZEISS), and the number of nuclei was counted using Imaris 8.1.2 (BITPLANE).

## Counting cell number in a single slice

Because the blastomere layers of zebrafish blastula were thicker than those of medaka blastula, it is difficult to detect DAPI signals from blastomeres located deep in zebrafish embryos. Therefore, we could not count the total cell number of zebrafish embryos using our systems. To compare total cell number per embryo in parallel with its estimation by qPCR, we counted the number of nuclei in single slice images obtained for immunofluorescence staining (Fig. EV5A).

## Estimation of post-fertilization cell cycle and cell cycle duration

In theory, the number of cells doubles per cell division. Therefore, Log2 (number of total cells per embryo measured in Fig. EV3A) was used as the number of post-fertilization cell cycles in medaka embryos (Fig. EV3B). The estimated number of cell cycles was further used to estimate the cell cycle duration (Fig. EV3C).

## Western blot

Dechorionated embryos were homogenized by gentile pipetting up and down in ice-cold PBS in microtubes. After centrifugation, the supernatant was removed, and the pellets were snap frozen in liquid nitrogen and stored in a freezer. Samples were boiled with Laemmli sample buffer at 95 °C for 5 min, run on SDS-polyacrylamide gels, and transferred to PVDF membranes (Immobilon-FL, Millipore, IPFL00010). Membranes were incubated with blocking buffer (Intercept blocking buffer, LI-COR, 927-60001) for 1 h at room temperature, incubated with primary antibody overnight at 4 °C (1/2000 × antibodies, 0.1% Tween-20 in blocking buffer), washed with TBST, incubated with secondary antibody (1/10,000 × IRDye, 0.1% Tween-20, 0.01% SDS in blocking buffer) for 1 h at room temperature, washed with TBST, and air dried for a few hours at room temperature in the dark. Imaging was performed with Odyssey CLx (LI-COR). Band identification and measurement of signal intensities were performed using Image Studio (LI-COR). Antibodies used for immunofluorescence are listed in Table EV2. Chameleon Duo Pre-stained Protein Ladder was used as the size marker (LI-COR, 928-60000).

For quantitative western blot, linearity of signal intensity was first confirmed using titrated samples as a standard (Figs. EV2F,G and EV5F,G). H3K9me3 intensity was then normalized by GAPDH intensity (Fig. 2E,F).

## RT-qPCR

Total RNA was extracted using ISOGEN (Nippon Gene) and RNeasy Mini kit (Qiagen). Total RNA was reverse transcribed using SuperSucript III (Invitrogen). qPCR was performed with the Mx3000P (Agilent) using the THUNDERBIRD SYBR qPCR mix (Toyobo). Average intensity of two technical replicates was used as expression level of one gene from each sample. To minimize the

difference in input cDNA amount, relative intensities of *sox2*, *zic2a*, and *tbx16* obtained by qPCR was normalized by that of *actb*. The primers used for qPCR are listed in Table EV1.

## RNA-seq analysis

First, the raw reads of medaka normal embryo RNA-seq were aligned to HdrR genome and processed as previously described (Nakamura et al, 2021). RPKM table was generated and the expression level was compared by DESeq2 (Love et al, 2014) (before ZGA: late morula vs after ZGA: late blastula), and genes upregulated after ZGA (adjusted *p*-value < 0.01, log2 fold change > 1.0) were used as zygotic genes in this study (Fig. EV2A). Second, the raw reads of medaka hybrid (female HNI × male HdrR) embryo RNA-seq were aligned to HdrR and HNI genome and processed as previously described (Nakamura et al, 2021). Reads uniquely mapped to HdrR, or paternal genome, were used to generate RPKM table by DESeq2. The RPKM was further normalized by total reads number (Fig. EV2A).

To show expression pattern of methyltransferases and demethylases, we processed previous RNA-seq data of medaka normal embryos (Nakamura et al, 2021; Ichikawa et al, 2017) as previously described (Nakamura et al, 2021), and RPKM table was generated by DESeq2.

## Statistical analysis

For two-sample statistical tests, normality and equal variances were first tested using the Shapiro-Wilk test and the F-test, respectively, with *p*-value = 0.05. If both null hypotheses were not rejected, a two-sided unpaired Student's t-test was performed. If the null hypothesis of F-test alone was rejected, we performed two-sided Welch's t-test. Otherwise, two-sided Wilcoxon rank-sum test was performed. $^{***}p < 0.001$, $^{**}p < 0.01$, $^{*}p < 0.05$, NS: not significant, respectively.

# Data availability

This study includes no data deposited in external repositories.

The source data of this paper are collected in the following database record: biostudies:S-SCDT-10_1038-S44319-024-00188-5.

# Peer review information

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

## Acknowledgements

We acknowledge all the laboratory members for everyday discussion and for their continuous support. We acknowledge Ryohei Nakamura (University of Tokyo) for critical reading of the manuscript and valuable discussion. This work was supported by JSPS KAKENHI Grant No. JP23K14121 to HSF, by JSPS KAKENHI Grant No. JP22K20625 and Grant No. JP23K14190 to TI, by the World-leading Innovative Graduate Program for Life Science and Technology, by the Ministry of Education, Culture, Sports, Science and Technology to SI, and by Japan Agency for Medical Research and Development (AMED) under Grant No. JP18gm1110007h0001 to HT.

## Author contributions

**Hiroto S Fukushima**: Conceptualization; Funding acquisition; Investigation; Methodology; Writing—original draft; Writing—review and editing. **Takafumi Ikeda**: Funding acquisition; Investigation. **Shinra Ikeda**: Methodology. **Hiroyuki Takeda**: Supervision; Funding acquisition; Writing—review and editing.

Source data underlying figure panels in this paper may have individual authorship assigned. Where available, figure panel/source data authorship is listed in the following database record: biostudies:S-SCDT-10_1038-S44319-024-00188-5.

## Disclosure and competing interests statement

The authors declare no competing interests.

# Expanded View Figures

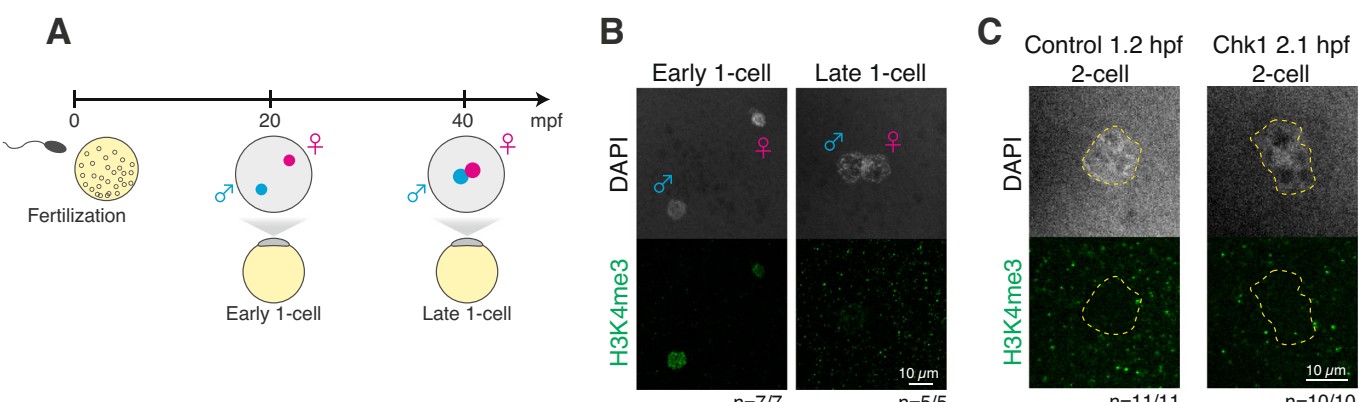

**Figure EV1. Supportive data for Fig. 1.**

(A) Development of medaka embryos at the one-cell stage. Blue and magenta indicate paternal and maternal pronuclei, respectively. mpf: minutes post fertilization. (B) Immunofluorescence staining of H3K4me3 at the one-cell stage. The number of embryos with the representative pattern is indicated at the bottom. (C) Immunofluorescence staining of H3K4me3 at the 2-cell stage in the chk1 experiment. Yellow dashed line indicates nuclei. The number of embryos with the representative pattern is indicated at the bottom.

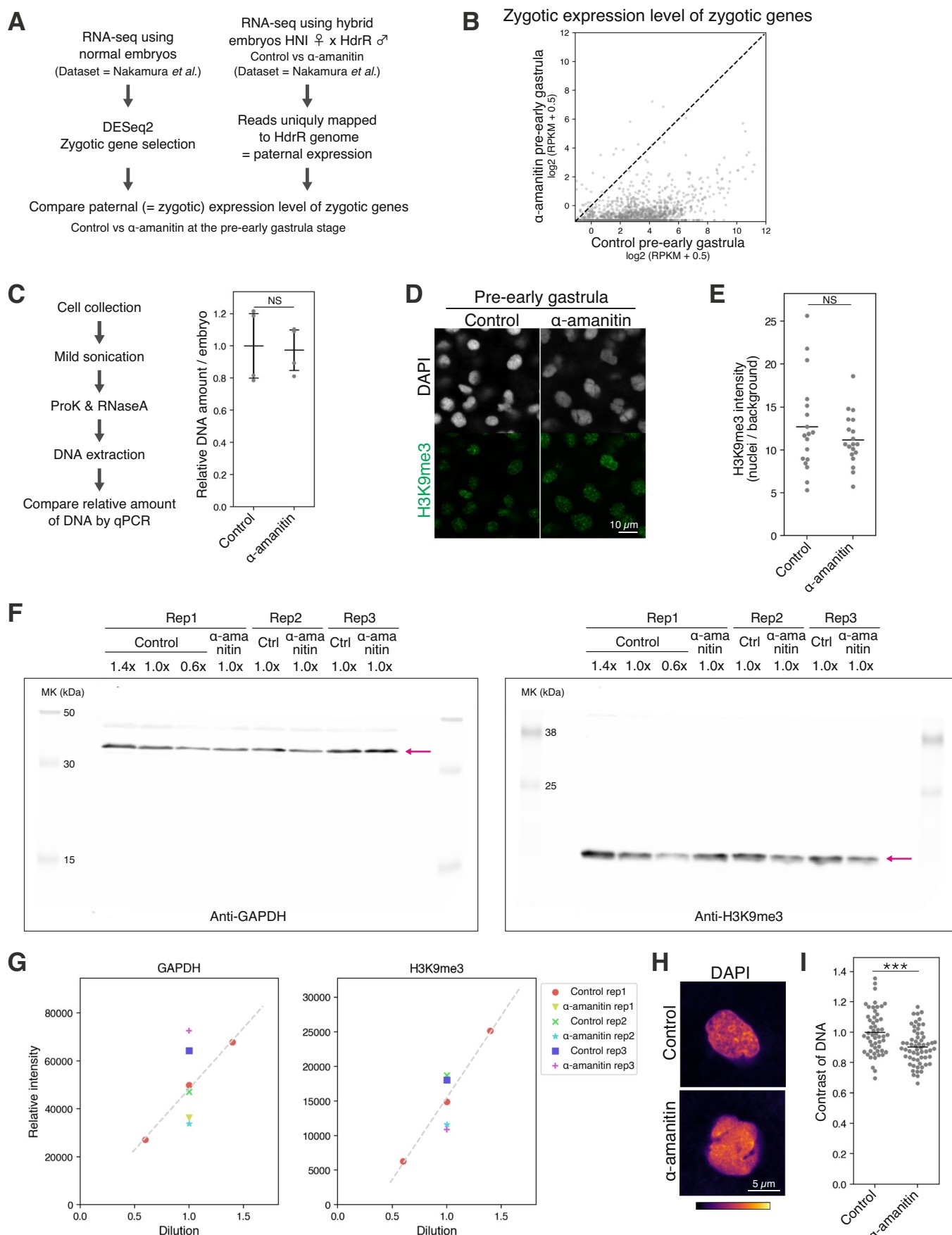

◀ **Figure EV2.  Supportive data for Fig. 2.**

(A) Procedure of analyzing zygotic expression level in α-amanitin-injected embryos using previous dataset (Nakamura et al, 2021). The number of RNA-seq biological replicate is $n = 2$ and 1 for normal embryos and hybrid embryos, respectively. See Methods for the detail. (B) Scatterplot indicates that α-amanitin injection impaired zygotic expression at the pre-early gastrula stage in medaka. Previous dataset (Nakamura et al, 2021) was analyzed as shown in (A) and Methods. (C) Procedure of quantification of relative amount of DNA per embryo (left) and the results (right) at the late blastula stage. Two-sided unpaired Student's t-test. Error bars indicate the mean ± s.d. $n = 4$ biological replicates. (D) Immunofluorescence staining of H3K9me3 in α-amanitin-injected medaka embryos at the pre-early gastrula stage. (E) Quantification of (D). Each dot indicates the average intensity of ~50 cells in a single broad field slice image of single embryo. Two-sided Welch's t-test. Bars indicate the means. $n = 18$ and 19 embryos for the control and α-amanitin, respectively. Data were pooled from two independent experiments. (F) Uncropped results of quantitative western blot at the late blastula stage using anti-GAPDH and anti-H3K9me3 antibodies. Magenta arrows indicate the specific bands. (G) Quantification of western blot signal intensities in Figs. 1F and EV2F. Scatter plots show that all signal intensities of western blots were within the linear range. (H) DAPI staining of control or α-amanitin-injected embryos at the late blastula stage. (I) Quantification of (H). Each dot indicates the DNA contrast of a single nucleus. Ten embryos were analyzed. Two-sided unpaired Student's t-test. Bars indicate the means. $n = 53$ and 59 nuclei for the late morula and late blastula, respectively. Data were pooled from two independent experiments. ***$p < 0.001$, NS: not significant.

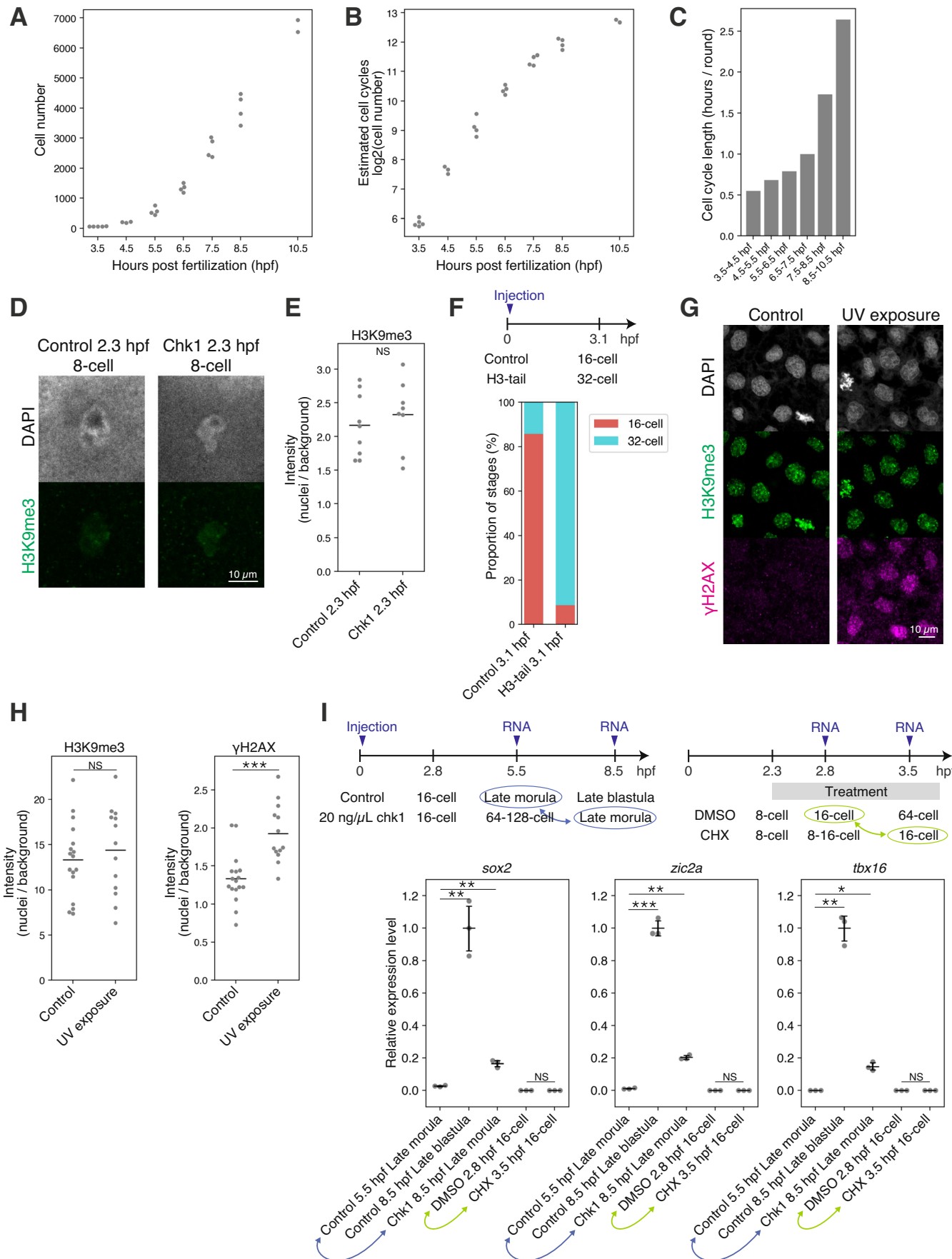

**Figure EV3. Supportive data for Fig. 3.**

(A) Number of cells per embryo at 3.5–10.5 hpf counted by Imaris software using DAPI staining data. $n = 5, 3, 4, 4, 4, 4, 2$ embryos for each stage. (B) Number of post fertilization cell-cycles estimated by total number of cells per embryo in (A). $n = 5, 3, 4, 4, 4, 4, 2$ embryos for each stage. (C) Cell cycle length estimated by cell cycle number and time line in (B). (D) Immunofluorescence staining of H3K9me3 in the chk1 injection experiment at the 8-cell stage (2.3 hpf). (E) Quantification of (D). Each dot indicates the average of single cells in a single broad field slice image of single embryo. Two-sided unpaired Student's t-test was performed. Bars indicate the means. $n = 9$ and 8 embryos for the control 2.3 hpf and chk1 2.3 hpf, respectively. Data were pooled from two independent experiments. (F) Schematic summarizing H3-tail injection (top) and proportion of stages of H3-tail-injected embryos in the cleavage stages (bottom). (G) Immunofluorescence staining of H3K9me3 and γH2AX in control and UV-treated embryos at the late blastula stage. (H) Quantification of (G). Each dot indicates the average of ~100 cells in a single broad field slice image of single embryo. Two-sided unpaired Student's t-test. Bars indicate the means. $n = 17$ and 13 embryos for the control and UV exposure, respectively. Data were pooled from two independent experiments. (I) Schematics summarizing RNA sampling (top) and RT-qPCR of zygotic genes (bottom). Expression level was first normalized by that of *actb* in each sample and subsequently normalized by the average expression level in control 8.5 hpf. Stages highlighted in blue or green were compared. Two-sided Welch's t-test. Error bars indicate the mean ± s.d. $n = 3$ biological replicates. *$p < 0.05$, **$p < 0.01$, ***$p < 0.001$, NS: not significant.

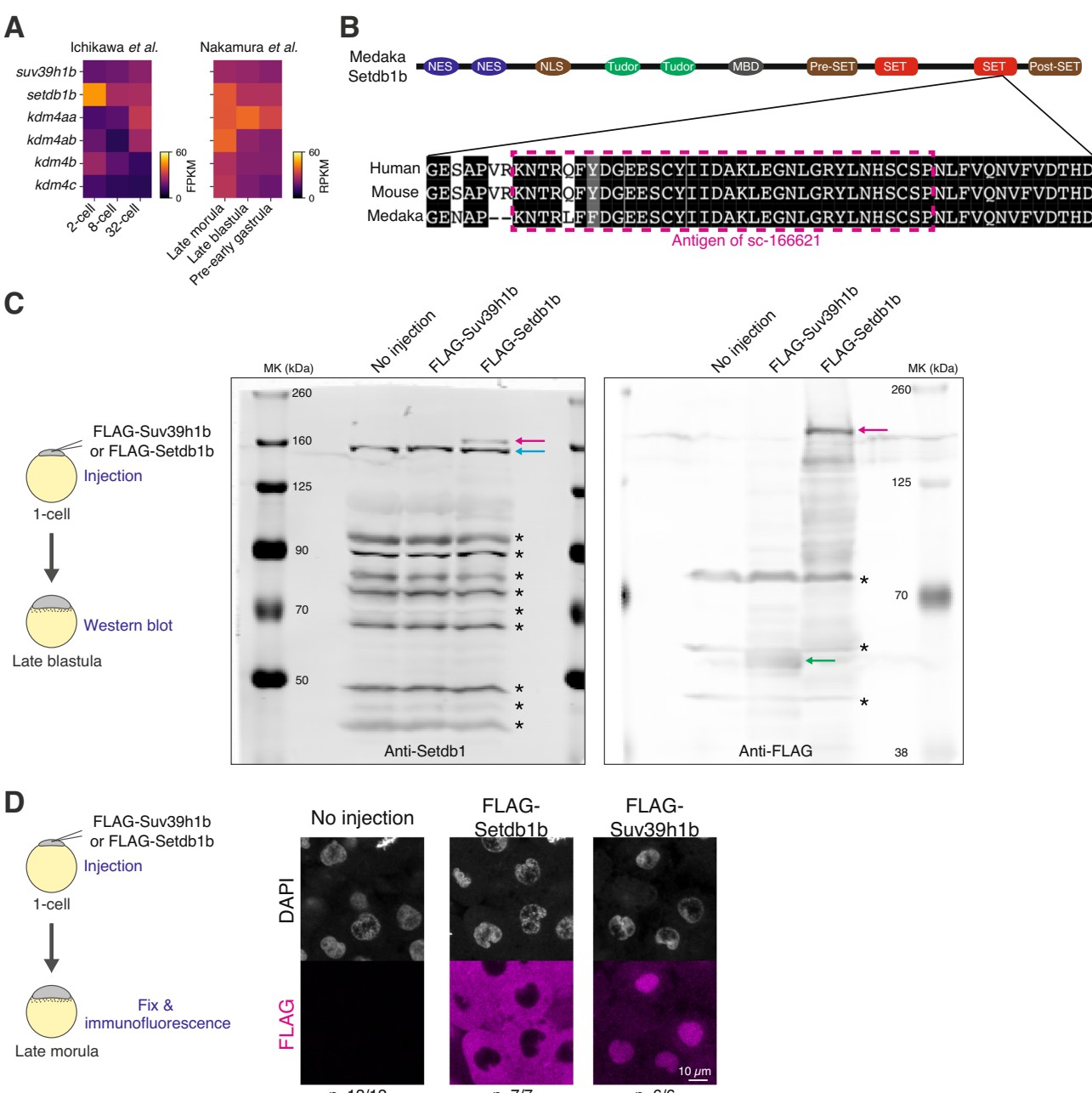

**Figure EV4. Supportive data for Fig. 4.**

(A) Expression level of H3K9me3 methyltransferases and demethylases during early development. Data was obtained from previous RNA-seq data (Nakamura et al, 2021; Ichikawa et al, 2017). (B) Domains of Setdb1 and amino acid sequence in the SET domain. The antigen sequence of the anti-Setdb1 antibody (sc-166621) is highlighted in magenta. (C) Schematic of the experiments to validate the specificity of the anti-Setdb1 antibody (sc-166621) (left) and the results of western blot at the late blastula stage (right). Blue, magenta, and green arrows indicate endogenous Setdb1b, exogenously expressed FLAG-Setdb1b, and exogenously expressed FLAG-Suv39h1b, respectively. Asterisks (*) indicate non-specific bands. (D) Schematic of the experiments to validate the localization of exogenously expressed FLAG-Suv39h1 and FLAG-Setdb1 (left) and immunofluorescence staining against anti-FLAG at the late morula stage (right). Consistent with the Fig. 4, exogenously overexpressed FLAG-Setdb1 localized to cytoplasm, while FLAG-Suv39h1 mainly accumulated in nuclei. The number of embryos with the representative pattern is indicated at the bottom.

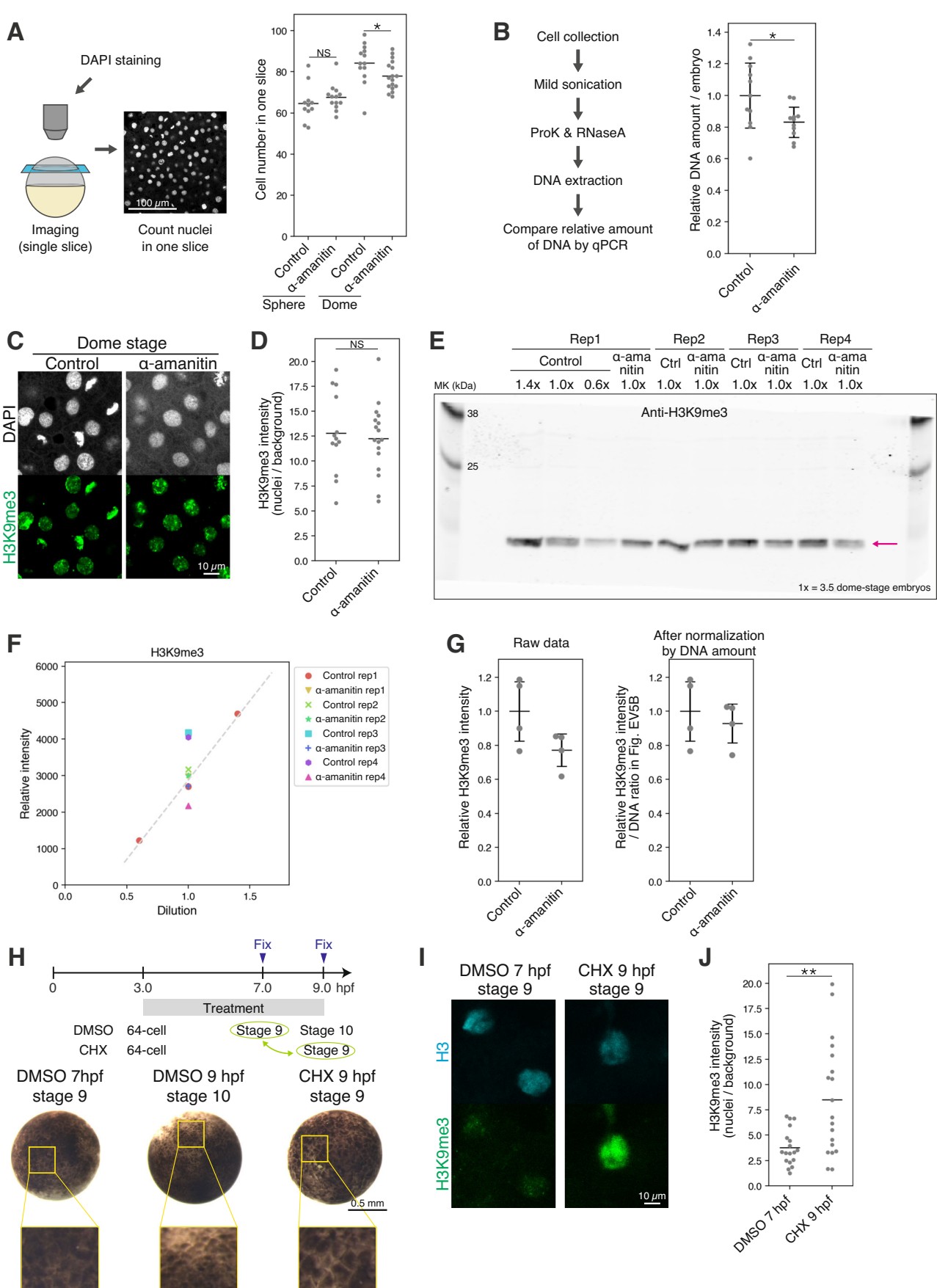

◀ **Figure EV5. Supportive data for Figs. 5 and 6.**

(A) Schematic of counting nuclei in an embryo using a single slice (left) and the cell number in α-amanitin-injected zebrafish embryos at the sphere or dome stage (right). Two-sided unpaired Student's t-test. Bars indicate the means. $n = 11$, 13, 13, and 17 embryos for the sphere control, sphere α-amanitin, dome control, and dome α-amanitin, respectively. Data were pooled from two independent experiments. (B) Procedure of quantification of the relative amount of DNA per embryo (left) and the results (right) in α-amanitin-injected zebrafish embryos at the dome stage. Two-sided Welch's t-test. Error bars indicate the mean ± s.d. $n = 11$ biological replicates. (C) Immunofluorescence staining of H3K9me3 in α-amanitin injection experiment at the dome stage. (D) Quantification of (C). Each dot indicates the average of ~80 cells in a single broad field slice image of single embryo. Two-sided unpaired Student's t-test. Bars indicate the means. $n = 13$ and 17 embryos for the control and α-amanitin, respectively. Data were pooled from two independent experiments. (E) Uncropped results of quantitative western blot at the dome stage using anti-H3K9me3 antibody. The Magenta arrow indicates the specific bands. The same number of dome-stage embryos (1× = ~3.5 embryos/lane) were loaded into each lane to compare total H3K9me3 levels per embryo. (F) Quantification of western blot signal intensities in (E). Scatter plots show that all western blot signal intensities were within the linear range. (G) Quantification of (E). On the right, data after normalization by the DNA ratio measured in Fig EV5B. Error bars indicate the mean ± s.d. $n = 4$ biological replicates. (H) Schematic summarizing the CHX treatment (top) and animal view of CHX-treated *X. laevis* embryos (bottom). Stages highlighted in green were compared in (I) and (J). To compare developmental stages, cells at the animal poles are magnified (yellow squares). (I) Immunofluorescence staining of H3 and H3K9me3 in CHX treatment at the stage 9. (J) Quantification of (I). Each dot indicates the average of ~5–10 cells in a single broad field slice image of single embryo. Two-sided Welch's t-test. Bars indicate the means. $n = 18$ and 19 embryos for the DMSO 7 hpf and CHX 9 hpf, respectively. Data were pooled from two independent experiments. *$p < 0.05$, **$p < 0.01$, NS: not significant.

