## [Peer Review File · EMBO Reports]

Cell cycle length governs heterochromatin reprogramming during early development in non-mammalian vertebrates

Hiroto Fukushima, Takafumi Ikeda, Shinra Ikeda, and Hiroyuki Takeda

Corresponding authors: Hiroto Fukushima (hiroto.fukushima@riken.jp), Hiroyuki Takeda (takeda_h@cc.kyoto-su.ac.jp)

Review Timeline:

Submission Date:	28th Jan 24
Editorial Decision:	26th Mar 24
Revision Received:	23rd Apr 24
Editorial Decision:	3rd Jun 24
Revision Received:	6th Jun 24
Accepted:	12th Jun 24

Editor: Esther Schnapp

Transaction Report:

Dear Dr. Fukushima,

Thank you for the submission of your manuscript to EMBO reports. We have now received the comments from 2 referees, which are pasted below. Unfortunately, referee 2 has not sent her/his report despite several reminders, and I am therefore making a decision on your ms now in order to save you from further loss of time.

As you will see, both referees agree and acknowledge that the study is interesting. They also suggest additional experiments to strengthen the findings, and I think all comments are good and should be addressed. Please let me know in case you disagree and we can discuss the revisions further, also in a video chat, if you like.

I would thus like to invite you to revise your manuscript with the understanding that the referee concerns must be fully addressed and their suggestions taken on board. Please address all referee concerns in a complete point-by-point response. Acceptance of the manuscript will depend on a positive outcome of a second round of review. It is EMBO reports policy to allow a single round of major revision only and acceptance or rejection of the manuscript will therefore depend on the completeness of your responses included in the next, final version of the manuscript.

We realize that it is difficult to revise to a specific deadline. In the interest of protecting the conceptual advance provided by the work, we recommend a revision within 3 months (26th Jun 2024). Please discuss the revision progress ahead of this time with the editor if you require more time to complete the revisions.

- 1) A data availability section providing access to data deposited in public databases is missing. If you have not deposited any data, please add a sentence to the data availability section that explains that.
- 2) Your manuscript contains statistics and error bars based on $n=2$. Please use scatter blots in these cases. No statistics should be calculated if $n=2$.

3) We replaced Supplementary Information with Expanded View (EV) Figures and Tables that are collapsible/expandable online. A maximum of 5 EV Figures can be typeset. EV Figures should be cited as 'Figure EV1, Figure EV2' etc... in the text and their respective legends should be included in the main text after the legends of regular figures.

5) a complete author checklist, which you can download from our author guidelines <https://www.embopress.org/page/journal/14693178/authorguide>. Please insert information in the checklist that is also reflected in the manuscript. The completed author checklist will also be part of the RPF.

6) Please note that all corresponding authors are required to supply an ORCID ID for their name upon submission of a revised manuscript (<https://orcid.org/>). Please find instructions on how to link your ORCID ID to your account in our manuscript tracking system in our Author guidelines <https://www.embopress.org/page/journal/14693178/authorguide#authorshipguidelines>

I look forward to seeing a revised form of your manuscript when it is ready. Please use this link to submit your revision:
<https://embor.msubmit.net/cgi-bin/main.plex>

Yours sincerely,

Referee #1:

The authors aimed to understand the mechanism by which heterochromatin marks such as H3K9me3 are erased and re-established in non-mammalian species such as medaka, zebrafish, and *Xenopus*. Through manipulating the speed of cell cycle using *chk1* mRNA overexpression, the authors suggested that H3K9me3 is passively removed before ZGA. Combining functional experiments with ZGA blocking and cell cycle control with *chk1*, the authors found that H3K9me3 establishment is attributed to cell cycle slowing but not ZGA. To elucidate which protein might be responsible for this, the authors focused on *Setdb1* and found its nuclear importing and exporting control is correlated with cell cycle speed. Finally, the authors observed similar results in zebrafish and *Xenopus* embryos, indicating that such mechanism is highly conserved among non-mammalian species.

Overall, the authors presented valuable insight into the dynamics of H3K9me3 and epigenetic reprogramming of heterochromatin during early embryonic development across multiple vertebrate model organisms. It is a very well written manuscript with elegant experiments design and clear description of the data, and the results are convincing and timely, considering the debate on the role of ZGA in H3K9me3 establishment (as illustrated in the paper). The paper could be further improved by additional controls and mechanistic/functional experiments related to *Setdb1*. Overall, I enjoyed the paper and support the paper's publication provided the authors can address the following comments.

Major comments:

1. It is a smart design to use *Chk1* overexpression to slow down cell cycle and analyze the erasure mechanism of H3K9me3. As the authors mentioned that H3K4me3 is probably actively erased, can the authors to perform similar experiments on H3K4me3 as a negative control to rule out the possibility that active erasure machinery might be affected by *chk1* overexpression?
2. The authors should be cautious when drawing the conclusion that DNA-dense regions are not heterochromatin based on the data that these regions are still there when H3K9me3 was removed, because H3K9me3 does not equal to heterochromatin. In mammals, whether DAPI-dense regions are still present when H3K9 methylation is absent is still debating.
3. The experiment result that cell cycle slowing can introduce H3K9me3 is quite informative. Can the authors examine development stages prior to cell cycle slowing to confirm that H3K9me3 is unaffected?
4. The correlation of *Setdb1* localization and cell cycle is intriguing. What is the expression patterns of *Setdb1* and *Suv39h1b* in medaka early development? Can the authors perform knockdown against *Setd1* to see if it is indeed responsible for H3K9me3 establishment? Any phenotype upon *Setdb1b* knockdown in medaka embryos?
5. How did *Setdb1b* enter the nuclei upon cell cycle slowing down? It would be helpful if the authors can at least speculate or discuss the underlying mechanisms.
6. While it is technically challenging, it would be extremely helpful if the authors can perform a few H3K9me3 ChIP-seq data/CUT&Run data in control and *Chk1* overexpressed embryos to make sure that the re-established H3K9me3 is indeed similar with prolonged cell cycle.
7. Can the authors show the expression patterns of H3K9me3 methyltransferase and demethylase in medaka embryo from the 1-cell to the pre-early gastrula stage?
8. Given that *chk1* overexpression and CHX treatment extended the cell cycle, were the time window for ZGA affected as well?

Minor comments:

1. For α -amanitin, it would be helpful to provide an RNA-seq analysis to confirm its effect.
2. In Figure EV4 B, right, why *Suv39h1b* was not detected when FLAG-*Suv39h1b* was overexpressed?
3. Was the concentration of *chk1* mRNA and α -amanitin injections titrated? For example, the injection concentration of α -amanitin in zebrafish seems to differ from that reported in Laue et al., 2019.
4. Can the authors explain/speculate why the prolonged cell cycles slightly increased the H3K9me3 level compared to that in 2-cell stage control embryos (Fig 1D-F, Control 1.2 hpf 2-cell vs *Chk1* 2.1 hpf)?
5. I wonder why cycloheximide (CHX) cannot be an alternative approach for cell cycle extending before 8-cell?
6. Would it be better if the embryos were treated with cycloheximide (CHX) starting from the 16-cell stage instead of 8-cell in Figure 3D? As described by the authors, the 16-cell stage is the time point when H3K9me3 is almost completely erased in medaka embryos (except for telomeres).
7. In Figure 6E, the morphology of *chk1* stage 9 embryos at 9hpf appears somewhat abnormal compared to stage 9 embryos at 7hpf in the control group. A more complete description of the *Chk1* overexpression effect on embryo development would be necessary, even though I think the main conclusion is likely not affected.
8. The embryonic stage and the number of replicates are missing in several figure legends.

Referee #3:

Review:

This manuscript supports a simple and global hypothesis that the resetting of repressive epigenetic marks (at least H3K9me3 marks) in externally developing metazoan animals occurs during the rapid embryonic nuclear cycles. It appears that the rapid divisions of eggs outpace the ability of a key histone methylase, SetDB1, to modify the replication added histones and the existing H3K9me3 marks are passively diluted out. Importantly, the developmental slowing of the cell cycles then triggers the onset of processes that reprogram the chromatin marks on the refreshed genome. The paper documents the dilution of H3K9me3 marks during the very earliest embryonic cycle in a fish model, the Japanese Killifish/medaka, and shows that slowing down these early cycles (by expression of the checkpoint kinase, Cdk1) allows persistence of the methylation. It also examines the later process as H3K9me3 marks begin to re-emerge as the cell cycle slows down. By injecting an inhibitor of transcription, α -amanitin, it is shown that this re-emergence is independent of zygotic transcription and events of the MBT. The findings are generalized, at least in part, to zebrafish and frog (*Xenopus Laevis*) and powerful parallels are drawn to earlier work in *Drosophila* and *C. elegans* to argue for a widespread role of this cell cycle coupled control of the re-setting of epigenetic marks.

The paper is not without some complications and weaknesses but overall makes a strong case.

The major complication/weakness is an apparent direct conflict with a previous study (Laue et al., 2019) that examined the onset of H3K3me3 in Zebrafish. While there are quite a number of differences between the studies, both examined whether inhibition of transcription prevented the re-emergence of H3K9me3, and Laue et al., report that it does and the present report shows that it doesn't. While the manuscript does acknowledge this discordance it doesn't provide a satisfying explanation of it. I do not see a flaw in the present report and think that the finding that an early arrest (64 cell stage) with cycloheximide treatment allows accumulation of H3K9me3 gives considerable confidence that slowing the cell cycle is sufficient without zygotic gene activity or MBT events (although the authors should be aware that blocking translation activates transcription in both fly and frog embryos - though probably an effect that is secondary to the block of the cell cycle - see Strong et al., 2020). I don't know if it is reasonable to demand that these authors reconcile the conflict with Laue et al., and I am not terribly bothered by it, but see suggestion below.

Less notable weaknesses are that some of the proposals in the paper are mostly founded on extensions of what was shown in previous work on other systems. For example, the relatively slow, but progressive accumulation, of SetDB1 following mitosis was shown *Drosophila* while in the data reported here the precise time during interphase is unknown and the imaging does not show the localization clearly. Additionally, I find the data and relevance of the results section "DNA-dense domains detected in nuclei after the MBT do not represent heterochromatin" unimpressive and their importance not clear (see Shermoen et al., 2010 for a more detailed treatment of the topic). The strength of this report is in analyzing three different models and conducting a series of simple experiments to show that H3K9me3 re-emergence depends on prolongation of the cell cycle and not on ZGA/MBT and showing that erasure at the beginning of development depends on rapid cycles in medaka.

Changes to presentation seem worthwhile. While the paper was quite understandable, I found some of the wording obtuse and it took me two or three tries to figure out what was being said. Additionally, I think it would be appropriate to more directly acknowledge when points are made by extension of findings by others. Finally, while I feel that the authors have documented their findings, it would be nice to have a more satisfying explanation of the discordance with results of Laue et al., and I offer the following as a possible explanation.

In *Drosophila* the earliest re-emergence of H3K9me3 is followed by a cascade of events that continues well into and beyond the MBT stage that fantastically amplifies the level of modification. So much so that in early analyses we overlooked the early events that nucleate the later steps. Judging by the blot in Laue et al., Fig. 1C right lane, the same is true in zebrafish. In flies, injection of α -amanitin causes an extra rapid cell cycle putting off MBT events for about 20 min, which I infer (not tested) would delay the rise in H3K9me3. Laue et al., used collections of 20+ embryos and blots rather than cytological detection (which use single visually staged embryos) to look at the effect of α -amanitin and they used 4.5 h embryos (I gather 30 min older than the embryos tested in this report). Given the huge rise in the blot signal in the later embryos, the signal they score might come from this later rise and α -amanitin might delay that rise (as we predict it would in flies) - this would show as an inhibition. It might be revealing to test if α -amanitin causes a delay in the later rise in modification - a rise which likely does depend on zygotic transcription directly or indirectly.

To all reviewers

We thank reviewers for their critical readings. Revised portions of the text are highlighted in the manuscript. To resolve editorial issues, the Fig EV6 was included to Fig EV5 in the revised manuscript. Below please find our detailed point-by-point responses to each comment.

Replies to comments from the reviewer #1

The authors aimed to understand the mechanism by which heterochromatin marks such as H3K9me3 are erased and re-established in non-mammalian species such as medaka, zebrafish, and *Xenopus*. Through manipulating the speed of cell cycle using *chk1* mRNA overexpression, the authors suggested that H3K9me3 is passively removed before ZGA. Combining functional experiments with ZGA blocking and cell cycle control with *chk1*, the authors found that H3K9me3 establishment is attributed to cell cycle slowing but not ZGA. To elucidate which protein might be responsible for this, the authors focused on *Setdb1* and found its nuclear importing and exporting control is correlated with cell cycle speed. Finally, the authors observed similar results in zebrafish and *Xenopus* embryos, indicating that such mechanism is highly conserved among non-mammalian species.

Overall, the authors presented valuable insight into the dynamics of H3K9me3 and epigenetic reprogramming of heterochromatin during early embryonic development across multiple vertebrate model organisms. It is a very well written manuscript with elegant experiments design and clear description of the data, and the results are convincing and timely, considering the debate on the role of ZGA in H3K9me3 establishment (as illustrated in the paper). The paper could be further improved by additional controls and mechanistic/functional experiments related to *Setdb1b*. Overall, I enjoyed the paper and support the paper's publication provided the authors can address the following comments.

Major comments:

1. It is a smart design to use *Chk1* overexpression to slow down cell cycle and analyze the erasure mechanism of H3K9me3. As the authors mentioned that H3K4me3 is probably actively erased, can the authors to perform similar experiments on H3K4me3 as a negative control to rule out the possibility that active erasure machinery might be affected by *chk1* overexpression?

> We thank the reviewer for the suggestion. We performed immunofluorescence staining of H3K4me3 using *Chk1*-overexpressed medaka embryos and confirmed that *Chk1* overexpression did not disturb rapid erasure of H3K4me3 (Fig EV1C, line

102-103).

2. The authors should be cautious when drawing the conclusion that DNA-dense regions are not heterochromatin based on the data that these regions are still there when H3K9me3 was removed, because H3K9me3 does not equal to heterochromatin. In mammals, whether DAPI-dense regions are still present when H3K9 methylation is absent is still debating.

> We thank the reviewer for the comment. We included the data because it could help in future understanding of how heterochromatin (not only the presence of H3K9me3 but also other features such as 3D structure, presence of HP1, transcriptional silencing, and so on) is established during early development. However, as the reviewer pointed out, our data was not sufficient to support our claim. We thus weakened our previous statement and modified our manuscript, accordingly (line 130-133).

3. The experiment result that cell cycle slowing can introduce H3K9me3 is quite informative. Can the authors examine development stages prior to cell cycle slowing to confirm that H3K9me3 is unaffected?

> We thank the reviewer for the suggestion. We performed immunofluorescence staining of H3K9me3 using 8-cell-stage embryos injected with *chk1* mRNA at the mild concentration, which was the stage before the cell cycle slowing became evident, and confirmed that the erasure of H3K9me3 was not affected (Fig EV3D, E). We revised our manuscript (line 148-149).

4. The correlation of *Setdb1* localization and cell cycle is intriguing. What is the expression patterns of *Setdb1b* and *Suv39h1b* in medaka early development? Can the authors perform knockdown against *Setd1* to see if it is indeed responsible for H3K9me3 establishment? Any phenotype upon *Setdb1b* knockdown in medaka embryos?

> We thank the reviewer for the suggestion. First, we show the expression patterns of *sedb1b* and *suv39h1b* using previous RNA-seq data (Ichikawa et al., 2017; Nakamura et al., 2021) (Fig EV4A). Both transcripts are present at all stages examined in this study. We revised our manuscript (line 190-191).

Second, we think that *Setdb1b* required for early development is mainly provided as a maternal protein. We found that cell cycle slowing during one-cell to 2-cell stages slightly but significantly increased H3K9me3 levels (Fig 1E, F) (also see the reply to this reviewer's minor comment #4). This indicates the presence of the writer (probably *Setdb1b*) even at the one-cell stage. Furthermore, we observed that H3K9me3 levels increased in the presence of CHX, a translation inhibitor, supporting

our idea that Setdb1 is maternally provided as a protein. Therefore, it is likely that morpholino-knockdown of Setdb1b would not work. We hope the reviewer understand this.

5. How did Setdb1b enter the nuclei upon cell cycle slowing down? It would be helpful if the authors can at least speculate or discuss the underlying mechanisms.

> We thank the reviewer for the suggestion. In the revised manuscript, we included our speculation on the mechanism of Setdb1b accumulation to nuclei upon cell cycle slowing (line 285-289).

6. While it is technically challenging, it would be extremely helpful if the authors can perform a few H3K9me3 ChIP-seq data/CUT&Run data in control and Chk1 overexpressed embryos to make sure that the re-established H3K9me3 is indeed similar with prolonged cell cycle.

> We agree with the reviewer opinion, but as the reviewer said, it is still technically difficult to do ChIP-seq or Cut&Run using Chk1-overexpressed embryos due to the limited number of cells isolated from cleavage-stage embryos. However, we think that at least we can discuss the global tendency of H3K9me3 dynamics during early stages with our current data.

7. Can the authors show the expression patterns of H3K9me3 methyltransferase and demethylase in medaka embryo from the 1-cell to the pre-early gastrula stage?

> We thank the reviewer for the suggestion. We provided the expression levels of H3K9me3 methyltransferases and demethylases during early development using previous RNA-seq data (Ichikawa et al., 2017; Nakamura et al., 2021) (Fig EV4A). Since all H3K9me3 methyltransferases and demethylases are present at all stages examined, transcriptional changes alone cannot explain the dynamics of H3K9me3 during early development. We revised our manuscript (line 190-191).

8. Given that chk1 overexpression and CHX treatment extended the cell cycle, were the time window for ZGA affected as well?

> We thank the reviewer for the suggestion. We performed RT-qPCR of some of zygotic genes in medaka and found that Chk1 overexpression modestly induced precocious initiation of ZGA at the late morula stage, but CHX treatment did not at all at the 16-cell stage (Fig EV3I). We revised our manuscript (line 176-181).

Minor comments:

1. For α -amanitin, it would be helpful to provide an RNA-seq analysis to confirm its effect.

> We thank the reviewer for the suggestion. We re-analyzed our previous RNA-seq data of hybrid medaka embryos (Nakamura et al., 2021) and showed that α -amanitin strongly inhibited zygotic expression from paternal alleles at the pre-early gastrula stage (Fig EV2A, B). We revised our manuscript (line 115).

2. In Figure EV4 B, right, why Suv39h1b was not detected when FLAG-Suv39h1b was overexpressed?

> We deeply apologize that our explanation was not sufficient in the original version. FLAG-Suv39h1b was detected by western blot using anti-FLAG antibody when FLAG-Suv39h1b was overexpressed, so we modified the figure and figure legend (green arrow in Fig EV4C).

3. Was the concentration of *chk1* mRNA and α -amanitin injections titrated? For example, the injection concentration of α -amanitin in zebrafish seems to differ from that reported in Laue et al., 2019.

> We deeply apologize for our miswriting on the experimental condition. We followed the previous protocol (Chan et al., 2019; Laue et al., 2019; Pálffy et al., 2020; Zhang et al., 2018) and injected 200 pg of α -amanitin into zebrafish one-cell stage embryo. We modified the manuscript (line 550).

As shown in Fig 1 and Fig 3, titration of *chk1* mRNA concentration affects the timing of initiation of cell cycle slowing. We did not titrate the concentration of α -amanitin, but we followed the previous protocol, and previous data showed the strong inhibition of ZGA by α -amanitin in medaka, zebrafish and *Xenopus* (Chan et al., 2019; Chen et al., 2019; Nakamura et al., 2021; Pálffy et al., 2020; Sudou et al., 2016; Zhang et al., 2018) (Fig EV2A, B). The developmental arrest observed in α -amanitin injection also supports strong inhibition of ZGA in our experiments (Fig 2B, 5D, 6D).

4. Can the authors explain/speculate why the prolonged cell cycles slightly increased the H3K9me3 level compared to that in 2-cell stage control embryos (Fig 1D-F, Control 1.2 hpf 2-cell vs Chk1 2.1 hpf)?

> We deeply apologize that our explanation was not sufficient in the original version. As we discussed, we speculate that this is due to the extension of cell cycle allowed the accumulation of Setdb1 in at the 2-cell stage (also see the response to this reviewer's major comment #4). We modified the manuscript (line 102, 276-277).

5. I wonder why cycloheximide (CHX) cannot be an alternative approach for cell cycle extending before 8-cell?

> We thank the reviewer for the comment. We agree that it is ideal to provide any alternative approaches for cell cycle slowing before the 8-cell stage, but there is a technical difficulty regarding the CHX treatment at early cleavage stages as follows. In general, we need to dechorionate medaka embryos for chemical exposure, but due to the hard chorion of medaka embryo, the dechorination is very time consuming. As a result, the 4-cell stage is the earliest stage at which we can start the chemical exposure using dechorionated medaka embryos. In case of H3K9me3, major erasure occurs by the 4-cell stage in medaka (Fig 1A-C). Therefore, the CHX exposure to dechorionated medaka early cleavage embryos cannot interfere the major reduction of H3K9me3. We hope the reviewer understand this technical difficulty.

6. Would it be better if the embryos were treated with cycloheximide (CHX) starting from the 16-cell stage instead of 8-cell in Figure 3D? As described by the authors, the 16-cell stage is the time point when H3K9me3 is almost completely erased in medaka embryos (except for telomeres).

> We thank the reviewer for the suggestion. Previously we thought that H3K9me3 was erased by the 16-cell stage (note that we did not examine the 8-cell stage in the previous study)(Fukushima et al., 2023). However, our immunofluorescence staining in the present study shows that there is no significant difference in global H3K9me3 level between the 8-cell stage and the 16-cell stage, suggesting that H3K9me3 is erased by the 8-cell stage. Therefore, we used both the 8-cell stage and the 16-cell stage as the stage with lowest H3K9me3 level, although we did not know if the retention of H3K9me3 is limited to telomeric regions at the 8-cell stage. To clarify our understanding more, we modified the manuscript (Line 84-85).

7. In Figure 6E, the morphology of *chk1* stage 9 embryos at 9hpf appears somewhat abnormal compared to stage 9 embryos at 7hpf in the control group. A more complete description of the Chk1 overexpression effect on embryo development would be necessary, even though I think the main conclusion is likely not affected.

> We thank the reviewer for the suggestion. As the reviewer pointed out, in *Xenopus* embryos, animal pole tends to be more sensitive to Chk1 overexpression than vegetable pole with unknown reason. We modified the manuscript and included the description of Chk1 overexpression phenotype in *Xenopus* embryos in more detail (line 257-258).

8. The embryonic stage and the number of replicates are missing in several figure legends.

> We are very sorry for our poor preparation of figures and legends. The figures and legends were revised as the reviewer suggested.

Replies to comments from the reviewer #3

Review:

This manuscript supports a simple and global hypothesis that the resetting of repressive epigenetic marks (at least H3K9me3 marks) in externally developing metazoan animals occurs during the rapid embryonic nuclear cycles. It appears that the rapid divisions of eggs outpace the ability of a key histone methylase, SetDB1, to modify the replication added histones and the existing H3K9me3 marks are passively diluted out. Importantly, the developmental slowing of the cell cycles then triggers the onset of processes that reprogram the chromatin marks on the refreshed genome. The paper documents the dilution of H3K9me3 marks during the very earliest embryonic cycle in a fish model, the Japanese Killifish/medaka, and shows that slowing down these early cycles (by expression of the checkpoint kinase, Cdk1) allows persistence of the methylation. It also examines the later process as H3K9me3 marks begin to re-emerge as the cell cycle slows down. By injecting an inhibitor of transcription, α -amanitin, it is shown that this re-emergence is independent of zygotic transcription and events of the MBT. The findings are generalized, at least in part, to zebrafish and frog (*Xenopus Laevis*) and powerful parallels are drawn to earlier work in *Drosophila* and *C. elegans* to argue for a widespread role of this cell cycle coupled control of the re-setting of epigenetic marks.

The paper is not without some complications and weaknesses but overall makes a strong case.

The major complication/weakness is an apparent direct conflict with a previous study (Laue et al., 2019) that examined the onset of H3K3me3 in Zebrafish. While there are quite a number of differences between the studies, both examined whether inhibition of transcription prevented the re-emergence of H3K9me3, and Laue et al., report that it does and the present report shows that it doesn't. While the manuscript does acknowledge this discordance it doesn't provide a satisfying explanation of it. I do not see a flaw in the present report and think that the finding that an early arrest (64 cell stage) with cycloheximide treatment allows accumulation of H3K9me3 gives considerable confidence that slowing the cell cycle is sufficient without zygotic gene activity or MBT events (although the authors should be aware that blocking translation activates transcription in both fly and frog embryos - though probably an effect that is secondary to the block of the cell cycle - see Strong et al., 2020). I don't know if it is reasonable to demand that these authors reconcile the conflict with Laue et al., and I am not terribly bothered by it, but see suggestion below.

> We thank the reviewer for the suggestion. First, we apologize for our poor

explanation on the difference between our present and the other previous studies (Laue et al., 2019), regarding the effect of α -amanitin injection on H3K9me3 re-accumulation in zebrafish. Please see our response in the final paragraph.

Second, regarding the effect of cell cycle manipulation of ZGA, we performed RT-qPCR of some zygotic genes in medaka and found that Chk1 overexpression modestly induced premature initiation of ZGA at the late morula stage, but CHX treatment did not at all at the 16-cell stage (Fig EV3I). We think this further exclude the possibility that re-accumulation of H3K9me3 depends on zygotic transcription. We revised our manuscript (line 176-181).

Less notable weaknesses are that some of the proposals in the paper are mostly founded on extensions of what was shown in previous work on other systems. For example, the relatively slow, but progressive accumulation, of SetDB1 following mitosis was shown *Drosophila* while in the data reported here the precise time during interphase is unknown and the imaging does not show the localization clearly. Additionally, I find the data and relevance of the results section "DNA-dense domains detected in nuclei after the MBT do not represent heterochromatin" unimpressive and their importance not clear (see Shermoen et al., 2010 for a more detailed treatment of the topic). The strength of this report is in analyzing three different models and conducting a series of simple experiments to show that H3K9me3 re-emergence depends on prolongation of the cell cycle and not on ZGA/MBT and showing that erasure at the beginning of development depends on rapid cycles in medaka.

> We thank the reviewer for the comment. First, we understand the weak and strong points of our study. However, our study is the first to extend the cell-cycle dependent mechanism to vertebrate models, medaka, zebrafish and *Xenopus*.

Second, we apologize our poor presentation on the section of DNA-dense domain. We included the data because it would help understanding on how heterochromatin (i.e. not only presence of H3K9me3 but also other features such as 3D structure, presence of HP1, transcriptional silencing, and so on) is established during early development. However, as the reviewer pointed, our explanation was not precise nor sufficient. We weakened our previous statement and modified our manuscript. We weakened our previous statement and modified our manuscript (line 130-133).

Changes to presentation seem worthwhile. While the paper was quite understandable, I found some of the wording obtuse and it took me two or three tries to figure out what was being said. Additionally, I think it would be appropriate to more directly acknowledge when points are

made by extension of findings by others. Finally, while I feel that the authors have documented their findings, it would be nice to have a more satisfying explanation of the discordance with results of Laue et al., and I offer the following as a possible explanation.

In *Drosophila* the earliest re-emergence of H3K9me3 is followed by a cascade of events that continues well into and beyond the MBT stage that fantastically amplifies the level of modification. So much so that in early analyses we overlooked the early events that nucleate the later steps. Judging by the blot in Laue et al., Fig. 1C right lane, the same is true in zebrafish. In flies, injection of α -amanitin causes an extra rapid cell cycle putting off MBT events for about 20 min, which I infer (not tested) would delay the rise in H3K9me3. Laue et al., used collections of 20+ embryos and blots rather than cytological detection (which use single visually staged embryos) to look at the effect of α -amanitin and they used 4.5 h embryos (I gather 30 min older than the embryos tested in this report). Given the huge rise in the blot signal in the later embryos, the signal they score might come from this later rise and α -amanitin might delay that rise (as we predict it would in flies) - this would show as an inhibition. It might be revealing to test if α -amanitin causes a delay in the later rise in modification - a rise which likely does depend on zygotic transcription directly or indirectly.

> We thank the reviewer for the suggestion. Having had the reviewer's comment, we re-analyzed our immunofluorescence data in medaka and zebrafish before and at early gastrulation stage and found that the single cell H3K9me3 levels do not dramatically increase during gastrulation (see Fig R1 below), suggesting that at the single cell level, H3K9me3 accumulation reaches a plateau by the blastula stage in fish embryos. In the re-analysis, we compared [late blastula from Fig 2D vs pre-early gastrula from Fig EV2E], and [sphere from Fig 5F vs dome from Fig EV5D]. These data were taken on the same day under the same experimental conditions. Together with our presented data, we would like to maintain our claim that the re-accumulation of H3K9me3 after the late morula proceeds independent of ZGA, and that the previous data showed an increase in H3K9me3 level per embryo (Laue et al., 2019) due to the difference in cell number per embryo. We do not think that it is necessary to include these data (Fig. R1) in the manuscript, because it is not our main argument. However, we revised our manuscript to make our opinion clearer (line 234-238).

Figure R1 Single-cell H3K9me3 levels did not dramatically increase upon gastrulation

(A) Quantification of immunofluorescence staining of H3K9me3 before (Late blastula) and at gastrulation (pre-early gastrula) in medaka. Each dot indicates the average intensity of ~40-50 cells in a single broad field slice image of single embryo. Two-sided unpaired Student's t-test. Error bars indicate the mean \pm s.d. Data were pooled from two independent experiments (biological replicates). Data are same as the control in Fig. 2D and Fig. EV2E.

(B) Quantification of immunofluorescence staining of H3K9me3 before (sphere) and at gastrulation (dome) in zebrafish. Each dot indicates the average intensity of ~60-80 cells in a single broad field slice image of single embryo. Two-sided Wilcoxon rank-sum test. Error bars indicate the mean \pm s.d. Data were pooled from two independent experiments (biological replicates). Data are same as the control in Fig. 5F and Fig. EV5D.

NS: not significant ($p > 0.05$).

References

- Chan, S.H., Tang, Y., Miao, L., Darwich-Codore, H., Vejnar, C.E., Beaudoin, J.-D., Musaev, D., Fernandez, J.P., Benitez, M.D.J., Bazzini, A.A., et al. (2019). Brd4 and P300 Confer Transcriptional Competency during Zygotic Genome Activation. *Dev. Cell* 49, 867-881.e8.
- Chen, H., Einstein, L.C., Little, S.C., and Good, M.C. (2019). Spatiotemporal Patterning of Zygotic Genome Activation in a Model Vertebrate Embryo. *Dev. Cell* 49, 852-866.e7.
- Fukushima, H.S., Takeda, H., and Nakamura, R. (2023). Incomplete erasure of histone marks during epigenetic reprogramming in medaka early development. *Genome Res.* 33.
- Ichikawa, K., Tomioka, S., Suzuki, Y., Nakamura, R., Doi, K., Yoshimura, J., Kumagai, M., Inoue, Y., Uchida, Y., Irie, N., et al. (2017). Centromere evolution and CpG methylation during vertebrate speciation. *Nat. Commun.* 8, 1833.
- Laue, K., Rajshekar, S., Courtney, A.J., Lewis, Z.A., and Goll, M.G. (2019). The maternal to zygotic transition regulates genome-wide heterochromatin establishment in the zebrafish embryo. *Nat. Commun.* 10, 1551.
- Nakamura, R., Motai, Y., Kumagai, M., Wike, C.L., Nishiyama, H., Nakatani, Y., Durand, N.C., Kondo, K., Kondo, T., Tsukahara, T., et al. (2021). CTCF looping is established during gastrulation in medaka embryos. *Genome Res.* 31, 968–980.
- Pálffy, M., Schulze, G., Valen, E., and Vastenhouw, N.L. (2020). Chromatin accessibility established by Pou5f3, Sox19b and Nanog primes genes for activity during zebrafish genome activation. *PLOS Genet.* 16, e1008546.
- Sudou, N., Garcés-Vásquez, A., López-Latorre, M.A., Taira, M., and del Pino, E.M. (2016). Transcription factors Mix1 and VegT, relocalization of vegt mRNA, and conserved endoderm and dorsal specification in frogs. *Proc. Natl. Acad. Sci.* 113, 5628–5633.
- Zhang, B., Wu, X., Zhang, W., Shen, W., Sun, Q., Liu, K., Zhang, Y., Wang, Q., Li, Y., Meng, A., et al. (2018). Widespread Enhancer Dememorization and Promoter Priming during Parental-to-Zygotic Transition. *Mol. Cell* 72, 673-686.e6.

Dear Dr. Fukushima,

Thank you for the submission of your revised manuscript. We have now received the enclosed reports from the referees, and I am happy to say that both support its publication now. We also like your cover suggestion very much.

Only a few editorial requests will need to be addressed before we can proceed with the official acceptance of your ms:

- Please correct the conflict of interest subheading to "Disclosure and Competing Interests Statement"
- Please remove the author credits from the ms file. All credits need to be entered during online ms submission.
- Some funding information is missing in our online submission system, please add.
- Please upload all main and all EV figures as individual Figure files.
- Please upload Tables EV1 and EV2 as separate excel files.
- All Source Data (SD) is zipped up in one folder, but the SD for each figure should be uploaded separately as one zipped folder per figure.
- The manuscript sections should be in the following order: Title page - Abstract & Keywords - Introduction - Results - Discussion - Methods - Data Availability - Acknowledgments - Disclosure Statement & Competing Interests - References - Figure Legends - Expanded View Figure Legends.
- There are several instances where n=2 biological replicates but statistics are calculated. This should not be done. Error bars and statistical tests should only be calculated for data that are derived from at least 3 independent experiments. For n=2 please show all datapoints along with their mean but no error bars. Best would be to repeat all experiments at least 3 times.
- Please note that in figures 2d, f, h, j; 4c-d; 5c, f, i; 6c, g; EV 2c, e, i; EV 3e, h-i; EV 5a-b, d, j; there is a mismatch between the annotated p values in the figure legend and the annotated p values in the figure file that needs to be corrected.

I would like to suggest some changes to the abstract that needs to be written in present tense. Please let me know whether you agree with this:

Heterochromatin marks such as H3K9me3 undergo global erasure and re-establishment after fertilization, and the proper reprogramming of H3K9me3 is essential for early development. Despite the widely conserved dynamics of heterochromatin reprogramming in invertebrates and non-mammalian vertebrates, previous studies have shown that the underlying mechanisms may differ between species. Here, we investigate the molecular mechanism of H3K9me3 dynamics in medaka (Japanese killifish, *Oryzias latipes*) as a non-mammalian vertebrate model, and show that rapid cell cycling during cleavage stages causes DNA replication-dependent passive erasure of H3K9me3. We also find that cell cycle slowing, toward the mid-blastula transition, permits increasing nuclear accumulation of the H3K9me3 histone methyltransferase Setdb1, and H3K9me3 re-accumulation. We further demonstrate that cell cycle length in early development also regulates (better: drives, governs, determines?) H3K9me3 reprogramming in zebrafish and *Xenopus laevis*. Together with the previous studies in invertebrates, we propose that a cell cycle length-dependent mechanism for both global erasure and re-accumulation of H3K9me3 is conserved among rapid-cleavage species of non-mammalian vertebrates and invertebrates such as *Drosophila*, *C. elegans* and teleost fish.

If possible, it would also be good to replace the word "regulates" in the title with a more specific term. What about "determines", "governs" or "drives", for example?

Cell cycle length drives heterochromatin reprogramming during early development in non-mammalian vertebrates

EMBO press papers are accompanied online by A) a short (1-2 sentences) summary of the findings and their significance, B) 2-3 bullet points highlighting key results and C) a synopsis image that is exactly 550 pixels wide and 200-600 pixels high (the height is variable). The synopsis image should provide a sketch of the major findings, like a graphical abstract. Please note that text needs to be readable at the final size. Please send us this information along with the final manuscript.

Referee #1:

The authors have performed thorough experiments and added proper speculation or discussion for their experiment results. I suggest that the profiling of H3K9me3 in Chk1-OE embryos is still worth trying in the future (not in this current manuscript), which may yield further insights. Nevertheless, I recommend this work to be published in the EMBO Reports.

Referee #3:

I felt that reviewer comments did not reveal any substantial deficiencies in the report and I am satisfied with the authors efforts to address what I feel are minor issues.

The authors addressed the minor editorial issues.

Dr. Hiroto Fukushima
RIKEN IMS
Suehiro-cho 1-7-22
Yokohama, Kanagawa 230-0045
Japan

Dear Dr. Fukushima,

I am very pleased to accept your manuscript for publication in the next available issue of EMBO reports. Thank you for your contribution to our journal.

Yours sincerely,
